# Glutamatergic basolateral amygdala to anterior insular cortex circuitry maintains rewarding contextual memory

Elvi Gil-Lievana[1], Israela Balderas[1], Perla Moreno-Castilla[1,4], Jorge Luis-Islas[2], Ross A. McDevitt[3], Fatuel Tecuapetla [1], Ranier Gutierrez [2], Antonello Bonci[4] & Federico Bermúdez-Rattoni[1✉]

Findings have shown that anterior insular cortex (aIC) lesions disrupt the maintenance of drug addiction, while imaging studies suggest that connections between amygdala and aIC participate in drug-seeking. However, the role of the BLA → aIC pathway in rewarding contextual memory has not been assessed. Using a cre-recombinase under the tyrosine hydroxylase (TH+) promoter mouse model to induce a real-time conditioned place preference (rtCPP), we show that photoactivation of TH+ neurons induced electrophysiological responses in VTA neurons, dopamine release and neuronal modulation in the aIC. Conversely, memory retrieval induced a strong release of glutamate, dopamine, and norepinephrine in the aIC. Only intra-aIC blockade of the glutamatergic N-methyl-D-aspartate receptor accelerated rtCPP extinction. Finally, photoinhibition of glutamatergic BLA → aIC pathway produced disinhibition of local circuits in the aIC, accelerating rtCPP extinction and impairing reinstatement. Thus, activity of the glutamatergic projection from the BLA to the aIC is critical for maintenance of rewarding contextual memory.

[1] División de Neurociencias, Instituto de Fisiología Celular, Universidad Nacional Autónoma de México, 04510 México City, Mexico. [2] Departamento de Farmacología, Centro de Estudios Avanzados, Instituto Politécnico Nacional, 07360 México City, Mexico. [3] Present address: Intramural Research Program, National Institute on Aging, National Institutes of Health, Baltimore, MD 21224, USA. [4] Present address: Global Institutes on Addiction, 1221 Brickell Ave, Miami FL33131, USA. ✉email: fbermude@ifc.unam.mx

The insular cortex (IC) is a region located in the temporal lobe of the brain that has been linked to the salience of stimuli[1,2], visceral control[3], recognition memory[4], and drug addiction[5,6]. It has been reported that human anterior IC (aIC) lesions completely suppress smoking addiction without relapse[7–10]. In addition, studies in rodents have demonstrated the participation of the aIC in the maintenance of drug addiction[11–16], although the cellular mechanisms underlying this process remain elusive.

Drugs of abuse are addictive due to the rewarding effects mediated by dopamine, a neurotransmitter released by ventral tegmental area (VTA) neurons that project to aIC and basolateral amygdala (BLA), among other areas[17–19]. However, it also has been postulated that the brain's reward circuitry is not only driven by dopaminergic neurotransmission; additional mechanisms, such as glutamatergic neurotransmission are involved in rewarding behavior[20,21]. Glutamatergic synaptic plasticity in addiction shares the same molecular mechanisms involved in learning and memory processes[20,22,23]. For example, tetanic stimulation of BLA induces long-term potentiation (LTP) in the IC, in an N-methyl-D-aspartate receptor (NMDAr)-dependent manner[24–26]. Moreover, the glutamatergic communication between the BLA and IC participates in the consolidation of several forms of memories like conditioned taste aversion[27,28]. Other studies have suggested that the BLA and the aIC are structures involved in rewarding behavior[29–32], and a magnetic resonance imaging study in humans has corroborated these findings[33]. However, the role of the BLA → aIC interaction in the maintenance and reinstatement of rewarding memory has not been described. This led us to hypothesize that the aIC glutamatergic afferents from the BLA are involved in the maintenance of rewarding contextual memory.

To this end, we optogenetically stimulated VTA dopamine neurons in a real-time conditioned place preference (rtCPP) paradigm, a behavioral model used to study the association of rewarding stimuli with contextual cues[34–37]. We found that optogenetic stimulation of tyrosine hydroxylase-expressing (TH+) VTA cell bodies produced electrophysiological responses in VTA neurons and mild but reliable modulation of aIC neuronal responses. In vivo microdialysis revealed a strong dopamine release in the aIC during VTA photoactivation. In addition, during rtCPP retrieval, we observed glutamate, dopamine and noradrenaline release in the aIC. Thus, we tested the role of these neurotransmitters during reconsolidation of the rtCPP memory. Several studies have suggested that consolidated memories may undergo a labile process during and after retrieval (updating), which destabilizes the original memory trace and makes it susceptible to being affected by amnestic agents, a process named reconsolidation[38,39]. Interestingly, disrupting the reconsolidation of memories associated with drugs of abuse has been identified as a potential clinical target for the treatment of addiction[40,41]. Accordingly, pharmacological manipulations revealed that only blockade of NMDA receptors in the aIC after the retrieval session accelerated rtCPP extinction and impaired rtCPP reinstatement. Moreover, photoinhibition of the glutamatergic BLA → aIC afferent fibers during retrieval accelerated rtCPP extinction and impaired subsequent reinstatement. Taken together, these results demonstrate that the glutamatergic BLA → aIC projection is necessary for the maintenance of rewarding contextual memories.

## Results

### Cortical modulation by photoactivation of VTA TH+ neurons.
To target midbrain dopamine neurons, we injected TH-cre mice bilaterally into VTA tissue with a virus encoding cre-dependent channelrhodopsin-2 (ChR2-eYFP). First, we characterized the expression of eYFP protein in VTA neurons and inputs from VTA into aIC of ChR2 mice to corroborate the presence of a dopaminergic VTA → aIC pathway. To this end, coronal VTA brain slices were analyzed. We observed that VTA neurons with endogenous tyrosine hydroxylase (TH) immunoreactivity were positive for eYFP near the viral injection site. Similarly, in coronal aIC slices, eYFP expression co-localized with TH+ terminals from VTA (Fig. 1a, b). Then, to characterize the functional modulation of the VTA → aIC circuit, we performed electrophysiological recordings in VTA or aIC in freely moving naïve mice during VTA TH+ neuron photoactivation. Figure 1c depicts a raster plot and peri-stimulus time histogram of a representative single neuron recorded in the VTA that increased its firing rate after each light pulse. This VTA neuron could follow frequencies up to 20 Hz. At higher frequencies and continuous stimulation, it exhibited an initial rapid peak followed by a gradual decay of the activity (Fig. 1c). The inhibitory component was more evident at continuous stimulation, suggesting that VTA TH+ neurons underwent strong negative control feedback at higher frequencies. The same modulatory pattern was observed at the population activity level (Fig. 1e). Briefly, 23.7% (111 out of 468) of neurons exhibited higher firing rates, whereas 15.3% (72/468) showed lower responses during, at least, in one laser frequency relative to activity in control trials (Fig. 1e, g). In contrast, aIC neurons responded with slight modulation during photoactivation of TH+ neurons in the VTA (Fig. 1d, f, h); 12.3% increased (48/390) whereas 12% decreased (47/390), which affected fewer aIC neurons than those modulated in VTA (Fig. 1h and Supplementary Fig. 1). In sum, electrophysiological recordings revealed that 20 Hz (5 ms width, 12–14 mW and 473 nm) was the optimal stimulation frequency for our preparation; therefore, we used it for all subsequent behavioral experiments.

### Activation of VTA TH+ neurons induces rtCPP.
To test the participation of the aIC in the extinction of rewarding contextual memory, we first characterized the rtCPP induced by optogenetic stimulation of VTA TH+ neurons (Fig. 2a). We found that ChR2 mice trained in a single 5- or 20-min session developed a mild place preference (Fig. 2b). However, ChR2 mice trained for three 20-min sessions on 3 consecutive days (one session per day) developed a strong conditioned preference for the side paired with optogenetic stimulation (Fig. 2c, e). The robust place preference was maintained for five continuous extinction sessions (one per day) until the conditioned preference disappeared completely. To establish whether a reinstatement stimulation could restore the rtCPP conditioning after almost complete extinction, photoactivation was applied to ChR2 mice during a reinstatement session (ChR2/with reinstatement) in the previously conditioned side following the acquisition protocol, but in a single 5 min session (Fig. 2d). Interestingly, the magnitude of place preference displayed by ChR2/with reinstatement mice in subsequent extinction sessions was much greater than that of ChR2/without reinstatement mice. These results show that once a contextual memory is extinguished, brief photoactivation of VTA TH+ neurons is sufficient to restore the preference for the conditioned side.

### rtCPP retrieval induces neurotransmitters release in the aIC.
We tested the neurochemical response in the aIC due to the photoactivation of the VTA in naïve unconditioned mice and, separately, as a consequence of the rtCPP memory retrieval (post-conditioning) session in previously conditioned mice. We used in vivo microdialysis in freely moving mice coupled with electrophoresis detection to determine the extracellular concentration of dopamine, noradrenaline, glutamate, and gamma-aminobutyric acid. During the photoactivation of VTA

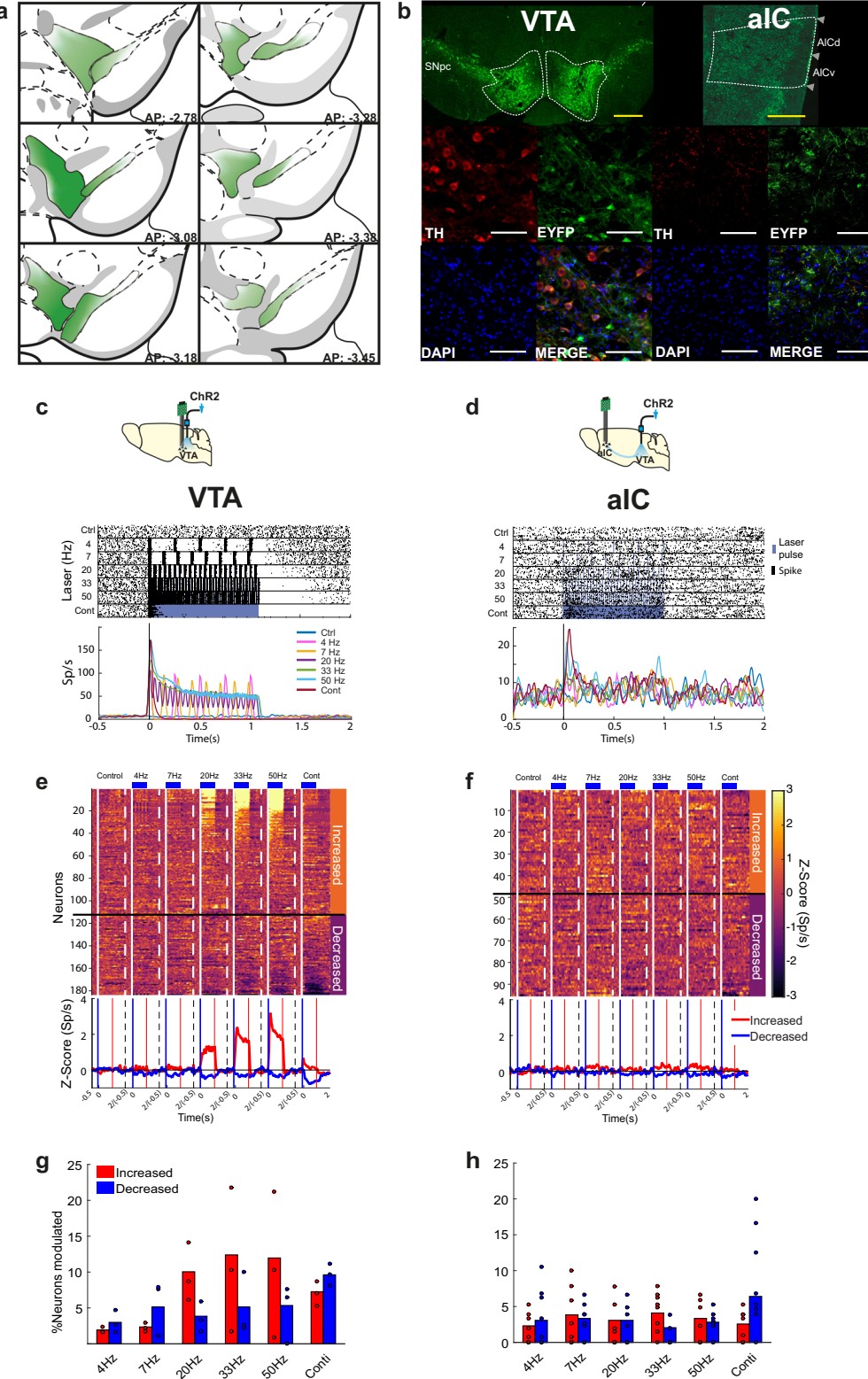

TH+ neurons in unconditioned mice (Fig. 3a), biochemical analysis revealed a strong increase of extracellular dopamine concentration (Fig. 3b). During the post-conditioning session, in the absence of the VTA TH+ neurons photoactivation (Fig. 3c), there was an increase of extracellular dopamine, noradrenaline, and glutamate concentrations (Fig. 3d). Dopamine was released during the photoactivation of the VTA TH+ neurons and during the post-conditioning session, whereas glutamate and

noradrenaline were only released during the post-conditioning session, indicating that these three neurotransmitters could be involved in the retrieval of a rewarding contextual memory.

**Cortical NMDA-receptor blockade accelerates rtCPP extinction.** Since the extracellular concentrations of dopamine, noradrenaline, and glutamate increased in the aIC during the post-rtCPP conditioning session, we administered intra-aIC injections

**Fig. 1 Photoactivation of VTA TH+ somas evokes a neuromodulatory response in the aIC. a** Diagram of virus infection and diffusion throughout anteroposterior extent of TH (red), 4′,6-diamidine-2′-phenylindole dihydrochloride (DAPI, blue), and eYFP (green) in somas at the VTA injection site (left panels) and in corresponding axons with VTA tissue. **b** Representative images of triple immunofluorescence in the aIC (right panels). Yellow scale bar represents 500 μm and white scale bar represents 100 μm. SNpc substantia nigra pars compacta, AICd agranular insular cortex dorsal, AICv agranular insular cortex ventral. **c** Raster plot and corresponding peri-stimulus time histogram (PSTH) (below, Sp/s, Spikes/s) of a VTA neuron recorded with an optrode (see scheme above). Responses were aligned to onset laser (time = 0); black ticks indicate a single action potential, whereas blue lines indicate laser pulses. **d** A representative neuron recorded in the aIC after photoactivation of VTA TH+ neurons (see scheme above). **e** Heat map of neuronal population responses recorded in VTA normalized to z-scores as a function of laser frequency. Neurons were sorted according to its modulatory pattern either increased or decreased firing rates relative to activity in control trials (Kruskal–Wallis test). Below, corresponding population PSTHs; red lines represent neurons with increased and blue with decreased responses in response to laser. **f** All neurons recorded with an electrode in aIC and photostimulation of VTA neurons. **g** Percentage of neurons modulated in the VTA during photostimulation of VTA somas at different frequencies. **h** Percentage of neurons modulated in aIC during photostimulation of VTA somas. In **g**, **h** each dot represents an individual mouse.

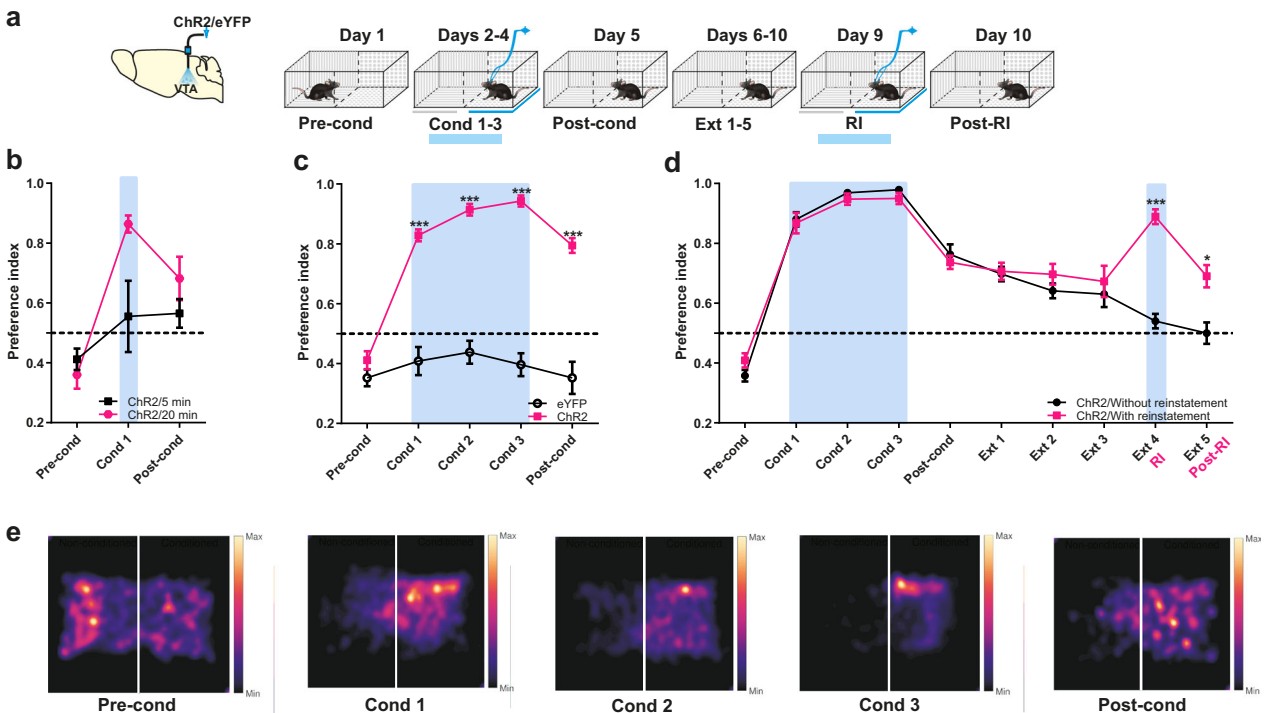

**Fig. 2 Photoactivation of VTA TH+ induces strong rtCPP. a** Diagram of virus infection, photoactivation of VTA TH+ neurons and time course of behavioral procedures. The preference of the compartments was determined during the pre-conditioning (Pre-cond). Optogenetic stimulation was performed in the least preferred place. Blue bars show photoactivation of VTA at 20 Hz (5 ms width), 12–14 mW and 473 nm. In the post-conditioning (Post-cond), preference of the compartments was tested after optogenetic stimulation conditioning. **b** Mice were conditioned during a single session (Cond 1) for 5 ($n = 5$) or 20 min ($n = 6$). Two-way repeated measures ANOVA with Greenhouse–Geisser correction factor showed no group effect: $F_{1,9} = 3.341$, $P = 0.1008$, time effect: $F_{1.237,11.14} = 15.47$, $P = 0.0015$, and interaction: $F_{2,18} = 4.480$, $P = 0.0264$. **c** Mice with ChR2 ($n = 15$) or eYFP ($n = 13$) expression in VTA were conditioned for three sessions (Cond 1-3) of 20 min. Two-way repeated measures ANOVA with Greenhouse–Geisser correction factor showed main group effect: $F_{1,26} = 187.8$, $P < 0.0001$, time effect: $F_{2.976,77.38} = 34.80$, $P < 0.0001$, and interaction effect: $F_{4,104} = 21.81$, $P < 0.0001$. Post hoc Bonferroni test eYFP vs. ChR2 for Cond 1-3 and Post-cond $P < 0.0001$. **d** Extinction rate for five sessions of 20 min of conditioning place preference (Ext 1-5) (ChR2/Without reinstatement group, $n = 8$) and place preference reinstatement (Post-RI) after 5 min optogenetic stimulation (RI) (ChR2/With reinstatement group, $n = 8$). Two-way repeated measures ANOVA with Greenhouse–Geisser correction factor showed main group effect: $F_{1,14} = 12.64$, $P = 0.0032$, time effect: $F_{4.852,67.93} = 79.80$, $P < 0.0001$, and interaction effect: $F_{9,126} = 9.385$, $P < 0.0001$. Post hoc Bonferroni test ChR2/Without reinstatement vs. ChR2/With reinstatement for Ext 4/RI $P < 0.0001$ and for Ext5/Post-RI $P = 0.0246$. **e** Representative heat maps of rtCPP phases. All data are shown as mean ± SEM. Dashed horizontal lines indicate no preference. *$P < 0.05$, ***$P < 0.0001$.

of antagonists for dopamine, noradrenaline, or glutamate receptors after rtCPP retrieval session (Fig. 4a). We observed that administration of the D1-like receptor antagonist SCH23390 or the β-adrenergic receptor antagonist propranolol did not affect maintenance of rtCPP (Fig. 4b). In contrast, intra-aIC injection of AP5, an NMDA-receptor antagonist, accelerated rtCPP extinction. Interestingly, rtCPP was re-established after a reinstatement session in the ChR2/vehicle group but not in the ChR2/AP5 group (Fig. 4c). In contrast, intra-aIC injections of cyanquixaline

(CNQX), a glutamate α-amino-3-hydroxy-5-methylisoxazole-4-propionic acid (AMPA) receptor antagonist, did not affect the maintenance of rtCPP (Fig. 4c). In order to know whether blockade of the NMDA receptors in the aIC after retrieval would have permanent nonspecific effects on rtCPP, we performed a reversal conditioning (Rcond 1–3) procedure. We observed that a previous blockade of NMDA receptors in the aIC did not impair the subsequent formation of new conditioning (Supplementary Fig. 2).

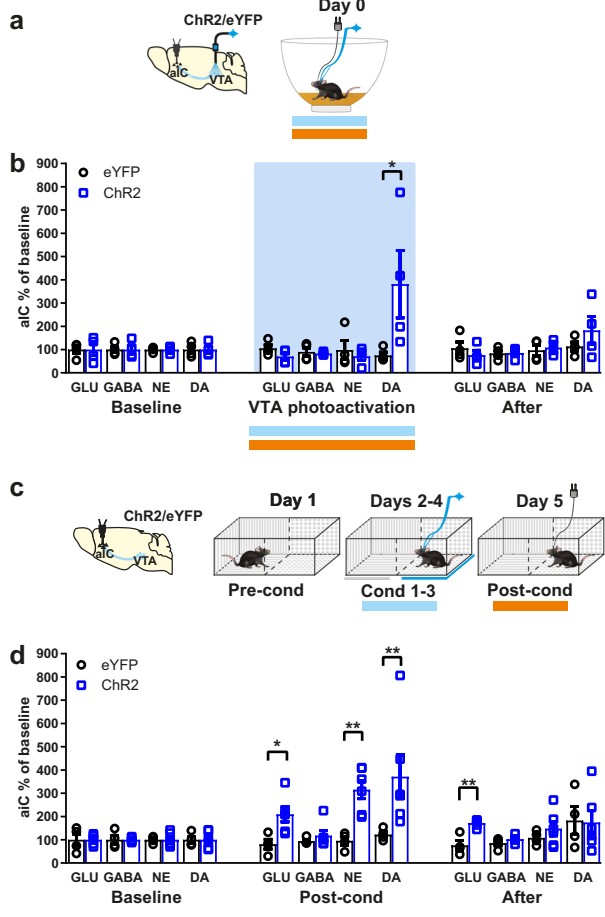

**Fig. 3 Photoactivation of VTA TH+ neurons induces dopamine release, whereas rtCPP retrieval induces dopamine, noradrenaline, and glutamate release in aIC. a** Diagram of virus infection, photoactivation of VTA TH+ neurons, and microdialysis guide cannula implantation into aIC. **b** Relative extracellular concentration of neurotransmitters compared with the baseline before, and after the photoactivation of VTA (blue bar) in eYFP (n = 4) and ChR2 (n = 4) mice. Horizontal blue bars indicate photoactivation of VTA. In vivo microdialysis (orange bar) revealed that extracellular concentration of dopamine (DA) increased during the VTA stimulation in ChR2 mice. For DA, Mann–Whitney test showed difference between eYFP vs. ChR2 during VTA stimulation $U$ value = 0, $P = 0.0286$. Noradrenaline (NA), Glutamate (GLU), and gamma-aminobutyric acid (GABA) did not show statistically significant changes. **c** Diagram of virus infection, microdialysis guide cannula implantation into aIC and real-time conditioning place preference (rtCPP). **d** Relative extracellular concentration of neurotransmitters compared with the baseline before, during (Post-cond) and after the rtCPP retrieval in eYFP (n = 4) and ChR2 (n = 6) mice. In vivo microdialysis (orange bar) revealed increased extracellular concentrations of DA, NE, and GLU due to the context exposure during rtCPP retrieval in ChR2 mice. For GLU, Mann–Whitney test showed difference between eYFP vs. ChR2 during Post-cond $U$ value = 1, $P = 0.0190$ and after rtCPP $U$ value = 0, $P = 0.0095$. For NA, Man–Whitney test showed difference between eYFP vs. ChR2 during Post-cond $U$ value = 0, $P = 0.0095$. For DA, Mann–Whitney test showed difference between eYFP vs. ChR2 during Post-cond $U$ value = 0, $P = 0.0095$. All data are shown as mean ± SEM. *$P < 0.05$, **$P < 0.001$.

**Inhibition of BLA → aIC pathway accelerates rtCPP extinction.**
Several studies have linked the stimulation of the BLA → IC pathway with plastic changes in the IC that are related to memory consolidation and extinction[42,43]. Similarly, high-frequency

stimulation of the BLA induces long-term facilitation in the IC and produces a more prolonged extinction of aversive memories[25]. In addition, microinjections of NMDA antagonists into the IC occlude changes in synaptic plasticity and long-term behavior[26,44–46]. Altogether, these data suggest that the fibers coming from the BLA to the IC are glutamatergic and could modulate memory consolidation.

To evaluate whether silencing glutamatergic afferent fibers from the BLA to the aIC could accelerate rtCPP extinction, we performed photoinhibition of the BLA → aIC fibers during or after rtCPP retrieval in ChR2/NpHR3.0 and ChR2/eYFP mice (Fig. 5a). We photo-stimulated aIC tissue with green light (12–14 mW, 532 nm) throughout the entire post-conditioning session, regardless of the side visited by the mouse. We found that inhibition of glutamatergic afferent inputs from BLA into aIC during the post-conditioning session accelerated the extinction process (Fig. 5b). Likewise, the CPP was restored after a reinstatement session in the ChR2/eYFP but not in the ChR2/NpHR3.0 group. However, when applied after termination of a post-conditioning session, photoinhibition did not affect the maintenance or extinction of rtCPP (Fig. 5c), suggesting the existence of a critical window where BLA → aIC inputs are necessary for maintenance of the rewarding contextual memory. Finally, in a different group of naïve ChR2/NpHR3.0 mice implanted with an optrode in aIC, we found that the somata of BLA neurons expressing NpHR3.0 send projection fibers to aIC (Fig. 6a, b). We then demonstrated that 1 min of continuous photoinhibition of glutamatergic BLA → aIC fibers resulted in a robust modulation of spontaneous activity in the aIC. Figure 6c, d shows two neurons recorded in aIC: one exhibiting excitation (panel c) and the other suppression (panel d) of neuronal responses evoked by green laser photoinhibition of BLA → aIC terminals. Around 48.5% (130/268) of aIC neurons exhibited an increased firing rate, whereas 19.8% (53/268) exhibited a reduced firing rate during photoinhibition of BLA → aIC glutamatergic afferent fibers (Fig. 6e, f). Thus, photoinhibition of glutamatergic BLA → aIC afferent fibers results in a disinhibition of local cortical circuits in the aIC. Similar results were obtained in mice that only expressed NpHR3.0 in BLA in the absence of ChR2 in VTA (Supplementary Fig. 3), suggesting a lack of crosstalk between opsins. All in all, these results highlight the importance of BLA glutamatergic modulation upon aIC activity in the maintenance of rewarding contextual memory.

## Discussion

There is extensive evidence that the mesolimbic dopaminergic system is involved in substance use disorders as well as in many forms of addiction. This system is mainly formed by dopaminergic neurons in the VTA that project to many other forebrain regions[47], including the aIC[17]. Stimulation of VTA dopaminergic neurons induces behavioral and cellular hallmarks of addiction, indicating sufficiency for the induction and progression of the disease[48]. Phasic stimulation produces action potentials of dopaminergic neurons in the VTA that are sufficient to establish a reliable rtCPP[34], used to measure the association of contextual reward[49]. Optogenetic stimulation of the VTA allows for precise control of the temporal and spatial activity of dopaminergic neurons. Consequently, by using this model we were able to examine the interaction of the VTA with other relevant brain circuits participating in maintenance and extinction of rtCPP. Our findings indicate that although the firing of VTA neurons clearly covaries with each photoactivation pulse in the same structure, one conditioning session under photoactivation was not sufficient to induce a reliable rtCPP. Three conditioning

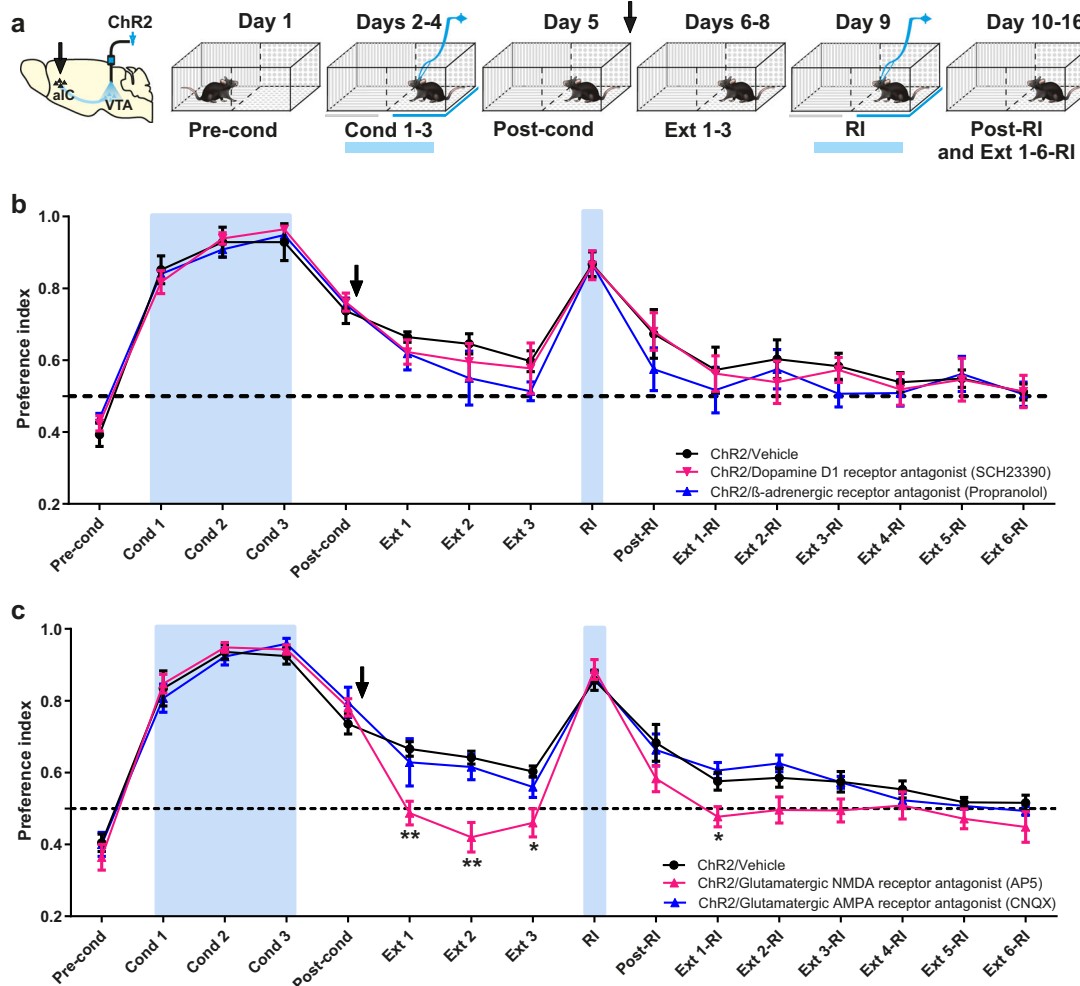

**Fig. 4 Post-conditioning microinjections of NMDA but not AMPA-receptor antagonist accelerate rtCPP extinction. a** Diagram of virus infection, photoactivation of VTA TH+ neurons, and timeline of behavioral procedures. The preference of the compartments was determined during the pre-conditioning (Pre-cond). Mice were conditioned during three sessions 20 min of photoactivation of VTA TH+ neurons (Cond 1-3). Blue bars show photoactivation of VTA. In the post-conditioning (Post-cond), preference of the compartments was tested after optogenetic stimulation conditioning. **b** Microinjections of dopamine D1 (SCH23390) and β-adrenergic receptor antagonist (propranolol) into aIC after post-conditioning did not affect the maintenance (Ext 1-3) nor the reinstatement (Post-RI) of the rtCPP induced by photoactivation of 5 min (RI) of the VTA. ChR2/vehicle ($n = 7$), ChR2/SCH23390 ($n = 7$), and ChR2/Propranolol ($n = 6$). Two-way repeated measures ANOVA with Greenhouse–Geisser correction factor showed no group effect: $F_{2,17} = 0.3844$, $P = 0.6866$, time effect: $F_{5.830,89.76} = 59.05$, $P < 0.0001$, and no interaction: $F_{30,255} = 0.5162$, $P = 0.9840$. **c** Microinjections of NMDA (AP5) but not the AMPA (CNQX) receptor antagonists into the aIC after post-conditioning accelerated rtCPP extinction (Ext 1-3) and blocked the reinstatement (Post-RI) of rtCPP (Ext 1-6-RI) induced by 5-min photoactivation of the VTA (RI). ChR2/vehicle ($n = 7$), ChR2/AP5 ($n = 7$), and ChR2/CNQX ($n = 7$). Two-way repeated measures ANOVA with Greenhouse–Geisser correction factor showed main group effect: $F_{2,18} = 9.099$, $P = 0.0019$, time effect: $F_{5.830, 104.9} = 103.7$, $P < 0.0001$, and interaction: $F_{30,270} = 2.216$, $P = 0.0005$. Post hoc Tukey test ChR2/Vehicle vs. ChR2/AP5 for Ext 1 $P = 0.0025$, Ext 2 $P = 0.0028$, Ext 3 $P = 0.0234$, Ext 1-RI $P = 0.0103$. All data are shown as mean ± SEM. The arrow indicates the time of administration of the drugs. Dashed horizontal lines indicate no preference. *$P < 0.05$, **$P < 0.01$.

sessions of 20 min with 20 Hz photoactivation of VTA TH+ neurons were required to produce a reliable and durable rtCPP. Likewise, we found that the maintenance of rtCPP was proportional to the duration of conditioning sessions. These findings give further support to the idea that memory stability relies on repeated exposure to stimuli[50]. In addition, our data uncovered that photoactivation of VTA TH+ neurons plays a weak neuromodulatory role in aIC neural responses in comparison with that of VTA stimulation. Accordingly, it has been reported that activation of the D1 receptor by itself does not elicit synaptic plasticity in rat hippocampus slices[51], whereas high-frequency-induced LTP is facilitated or impaired by agonists or antagonist of dopamine receptors in the hippocampus and/or prefrontal cortex[52–55].

Using in vivo microdialysis, we observed that re-exposure of mice to the conditioning context during a post-conditioning session induced strong release of dopamine, noradrenaline, and glutamate. These results suggest that exposure to the conditioned context during retrieval could be the triggering factor for the release of glutamate, dopamine, and noradrenaline in the aIC. In this regard, findings obtained in experiments using other aIC-dependent behavioral tasks suggest that the release of catecholamines and glutamate during retrieval is involved in the reconsolidation and maintenance of the conditioned responses[28,56].

The phenomenon of reconsolidation posits that retrieval induces memory destabilization, creating a window in which consolidated memories are susceptible to pharmacological interference or updating with new information[39,57]. In this way, after

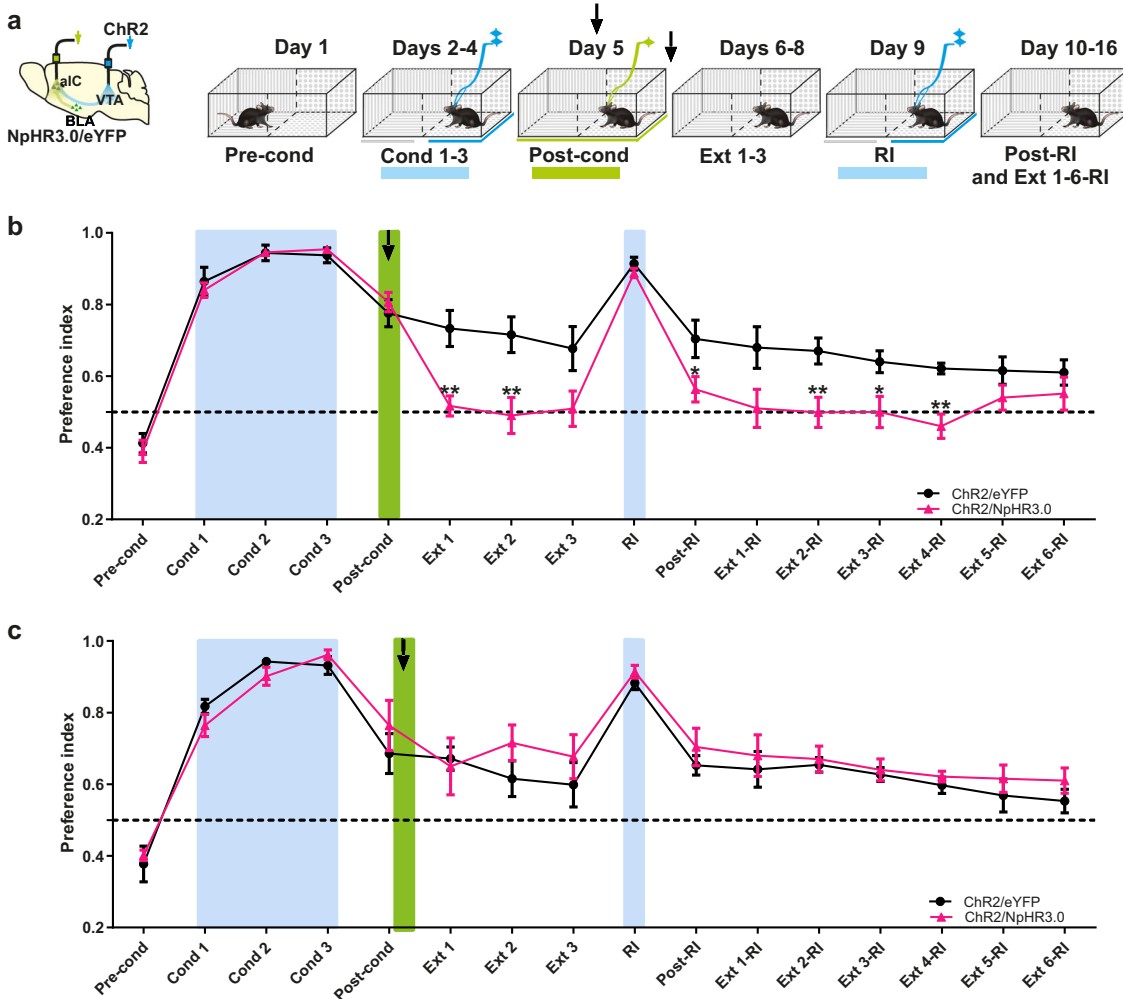

**Fig. 5 Photoinhibition of inputs from the BLA to the aIC accelerate rtCPP extinction. a** Diagram of virus infection into VTA TH+ neurons, photoinhibition of BLA neurons, and timeline of behavioral procedures. The preference of the compartments was determined during the pre-conditioning (Pre-cond). Mice were conditioned during three sessions 20 min of photoactivation of VTA TH+ neurons (Cond 1-3). Blue bars indicate photoactivation of VTA; green bar indicates photoinhibition of aIC. **b** Photoinhibition of the aIC during the post-conditioning (Post-cond) accelerated extinction in the ChR2/NpHR3.0 ($n = 9$) but not in the ChR2/eYFP ($n = 7$) group. Green bar shows photoinhibition of the aIC. The green laser was constantly pulsed at 12–14 mW and 532 nm. Two-way repeated measures ANOVA with Greenhouse–Geisser correction factor showed main group effect: $F_{1,14} = 7.266$, $P = 0.0174$, time effect: $F_{4.704,65.85} = 62.50$, $P < 0.0001$, and interaction: $F_{15,210} = 4.406$, $P < 0.0001$. Post hoc Bonferroni ChR2/eYFP vs. ChR2/NpHR3.0 for Ext 1 $P = 0.0042$, Ext 2 $P = 0.0068$, Post-RI $P = 0.0483$, Ext 2-RI $P = 0.0082$, Ext 3-RI $P = 0.0206$, Ext 4-RI $P = 0.0013$. **c** Photoinhibition of the aIC after post-cond did not affect the maintenance of the rtCPP in any of the two groups, NpHR3.0 ($n = 7$) and eYFP ($n = 7$). Two-way repeated measures ANOVA with Greenhouse–Geisser correction factor showed no group effect: $F_{1,12} = 1.178$, $P = 0.229$, time effect: $F_{5.815,69.78} = 29.01$, $P < 0.0001$, and no interaction: $F_{15,180} = 0.6331$, $P = 0.8450$. All data are shown as mean ± SEM. The arrow indicates the time of inhibition of BLA → aIC circuitry. Dashed horizontal lines indicate no preference. *$P < 0.05$, **$P < 0.001$.

retrieval, new information is compared with the already stored memories, which can thus be destabilized. Subsequently, both the destabilized memory and the incoming information are re-stabilized or reconsolidated into an updated long-term memory[39,58]. In the rtCPP model, when mice are re-exposed to the learning environment a labile conditioned memory could be affected by amnesic drugs such as NMDA antagonists, impairing previous memory, and accelerating rtCPP extinction. Such results have been reported with fear conditioning, object recognition memory, and traditional CPP[59–61]. We found that NMDA receptors are critical for maintaining rtCPP, whereas D1, β-adrenergic, and AMPA receptors do not appear to participate in the maintenance of rtCPP memory. Blockade of NMDA receptors in the aIC may destabilize the original memory trace, thereby accelerating extinction and impairing reinstatement to allow the acquisition of new reinforcement learning. This possibility is

supported by evidence showing that reconsolidation mechanisms in recognition memory require the activation of NMDA but not AMPA receptors[59]. Our results further demonstrate that intra-aIC administration of NMDA-receptor antagonists produced a reliable and long-lasting impairment of rtCPP maintenance. Therefore, it is possible that the aIC could be an important hub that links contextual and visceral information related to the maintenance of memories associated with addictive drugs[13,62].

Findings of several experiments suggest important communication between the amygdala and the IC during appetitive and aversive memory formation[26,62–64]. For example, a reliable LTP in the aIC was induced by high-frequency stimulation in the BLA, which was blocked by intracortical administration of an NMDA antagonist[24,25,45]. These results suggest that cortical plasticity induced by amygdala stimulation could be regulated by glutamate through activation of NMDA receptors in the aIC. Therefore, we

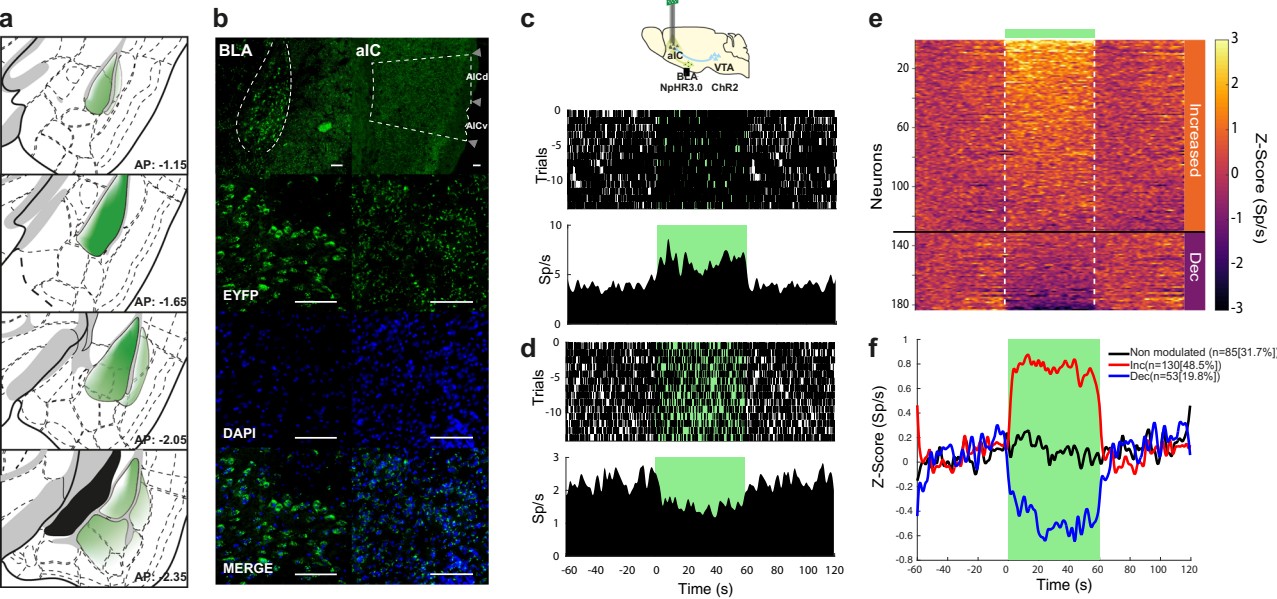

**Fig. 6 Continuous photoinhibition of glutamatergic BLA → aIC fibers provoke disinhibition of local cortical circuits in the aIC. a** Diagram of virus infection and diffusion throughout BLA anteroposterior axis. **b** Representative images of double immunofluorescence for eYFP (green) and 4′,6-diamidine-2′-phenylindole dihydrochloride (DAPI; blue) in basolateral amygdala (BLA) somas (left panels) and anterior insular cortex (aIC) axons (right panels). White scale bar represents 100 μm. AICd agranular insular cortex dorsal, AICv agranular insular cortex ventral. **c** Raster plot and corresponding PSTH (below) of a representative excited neuron recorded in aIC. Responses were aligned (time = 0) to onset of photoinhibition for 1 min (shown by green rectangle). Black ticks indicate action potentials. Top, scheme of the experimental protocol. **d** Representative aIC neuron exhibiting suppression during photoinhibition. **e** Z-score normalized color-coded PSTH of neurons with either increased (Inc) or decreased (Dec) evoked responses recorded in the aIC while the glutamatergic BLA → aIC fibers were opto-inhibited. Red color indicates higher firing rates, whereas blue color inhibitory responses. Lines depict the start and the end of photoinhibition of green laser. **f** Mean Z-scores of neuronal populations categorized by significant increase (inc), decrease (dec), or no effect during photoinhibition.

hypothesized that BLA → aIC pathway could be involved in the maintenance of a rewarding contextual memory. We found that the photoinhibition of BLA projections to the aIC during rtCPP retrieval accelerated the extinction process of a rewarding contextual memory. However, the inhibition of BLA projections to the aIC post-retrieval did not accelerate rtCPP extinction. This effect is time-dependent, because it destabilizes the original memory trace only when the memory is in a labile state and susceptible to interference. We interpret these results to mean that optogenetic inhibition of BLA projections to the aIC impairs reconsolidation of rewarding contextual memory, but we cannot rule out that other alternative mechanisms are involved. The cellular mechanisms mediating the effects of photoinhibition of glutamatergic terminals and NMDA-receptor antagonists remain to be determined. It is possible that these two interventions unleash similar molecular events. These findings support the idea that glutamatergic signaling in the aIC constitutes an important mechanism for the maintenance and the re-establishment of rewarding contextual memory.

In summary, by employing optogenetic and behavioral approaches, we report that inhibition of the BLA → aIC pathway accelerates the extinction and impairs the re-establishment of a rewarding contextual memory. We observed a similar behavioral pattern by the pharmacological blockade of NMDA receptors in the aIC. Our data strongly suggest that the BLA → IC pathway is centrally involved in the maintenance of rewarding contextual memory through NMDAR-dependent glutamatergic signaling.

## Methods

**Animals.** TH-Cre mice (TH, FI12 line) express Cre-recombinase protein under control of the endogenous TH promoter. Breeder mice were kindly donated by Dr Rui M. Costa from the Champalimaud (Center for the Unknown) and crossed onto C57BL/6J mice for at least six generations. Male and female experimental TH-Cre mice were 3 to 4 months old (23–35 g body weight) at the initiation of

experiments. No differences were found between male and female TH-Cre mice in any of the experiments performed. Mice were individually housed with food and water available ad libitum in a light/dark cycle room (12:12 h) with temperature of 22 ± 2 °C and relative humidity of 50 ± 5%. The behavioral tests were carried out during the light phase of the cycle. All procedures were approved by the Instituto de Fisiología Celular (FBR125-18) Institutional Animal Care and Use Committee and complied with the guidelines established in the Official Mexican Standard (NOM-062-ZOO-1999).

**Genotyping.** We used the HotSHOT method[65] for DNA extraction. The method consists of the lysis of 1 mm of a tail snip in an alkaline reagent (NaOH 25 mM, diodium EDTA 0.2 mM) under heat conditions (95 °C, 1 h) and further neutralization with a suitable buffer (Tris-HCl 1 M at pH 7.5). After centrifugation at 2500 rpm by 2 min, the supernatant containing the DNA was recovered. This DNA was used for amplification through polymerase chain reaction (PCR) with the QIAGEN kit. The primers used for PCR were Cre Reverse 5′GGTTTCCCGCA GAACCTGAA and Cre Forward 5′AGCCTGTTTTGCACGTTCACC.

**Viral vector.** The Cre-inducible adeno-associated virus (AAV) was obtained from the University of North Carolina (UNC) Gene Therapy Center Vector Core. The viral concentration was $5.2 \times 10^{12}$ virus molecules ml$^{-1}$ for rAAV5/EfIα-DIO-hChR2(H134R)-eYFP (ChR2), $6.0 \times 10^{12}$ virus molecules ml$^{-1}$ for rAAV5/EfIα-DIO-eYFP (eYFP), $5.2 \times 10^{12}$ virus molecules ml$^{-1}$ for rAAV5-CamKIIa-eEYFPeYFP (eNpHR3.0), and $5.1 \times 10^{12}$ virus molecules ml$^{-1}$ for rAAV5-CamkIIa-eYFP (eYFP). Viruses were subdivided into aliquots stored at −80 °C until use.

**Stereotaxic surgery.** Mice were anesthetized with isoflurane (induction 3%, maintenance 1–1.5%, ViP 3000 Matrix). The microinjection needles (29-G) were connected to a 10 μl Hamilton syringe and filled with AAV. For all experiments, the mice were bilaterally injected with AAV (0.5 μl) at a rate of 0.1 μl min$^{-1}$ with additional 5 min for a complete diffusion and implanted with 200 μm core optic fibers through 1.25-mm-wide zirconia ferrules in each hemisphere. The ChR2 or eYFP vectors were injected into the VTA (from Bregma (mm): −3.08 AP, ±0.60 ML, and −4.80 DV. The NpHR3.0 or eYFP vectors were injected into the BLA (from Bregma (mm): −2.05 AP, ±3.40 ML, and −5.30 DV). The optic fibers were implanted above the VTA (from Bregma (mm): −3.08 AP, ±1.20 ML, and −4.30 DV at 10° angle) or the aIC (1.70 AP, ±2.60 ML, and −2.75 DV). For microdialysis experiments, the mice were implanted unilaterally with a guide cannula (CMA/7:

CMA Microdialysis) aimed at the aIC (from Bregma (mm): +1.40 AP, ±3.30 ML, and −3.5 DV). For pharmacological experiments, the mice were implanted with bilateral guide cannulas (C316G-24 ga, Plastics One) into the IC (from Bregma (mm): +1.40 AP, ±3.30 ML, and −3.0 DV). Coordinates were taken from Allen's reference atlas of the mouse brain[66].

**General procedures**. The behavioral experiments began 3–4 weeks after surgery for the groups injected in the VTA to allow sufficient expression of opsins. They received optogenetic stimulation in the VTA or aIC through optical tethers, which consisted of a diode-pumped solid-state blue (473 nm) and green (532 nm) laser (150 mW; OEM Laser Systems) coupled to 62.5 μm core, 0.22 NA standard multimode hard-cladding optical fiber (Thorlabs) that passed through a single-channel optical rotary joint (Doric Lenses) prior to being split 50:50 with a fused optical coupler. The intensity of light output was 12–15 mW per split fiber for all experiments. The mice were connected to the laser just before starting each behavioral or microdialysis session. For pharmalogical studies, DL-2-amino-5-phosphonopentanoic acid (10 mg ml$^{-1}$, DL-AP5, #0105 Tocris Bioscience), an N-methyl-D-aspartate (NMDA) receptor antagonist; cyanquixaline (1 mg ml$^{-1}$, CNQX disodium salt hydrate, #C239 Sigma-Aldrich), an α-amino-3-hydroxy-5-methylisoxazole-4-propionic acid (AMPA) receptor antagonist; halobenzazepine (2 mg ml$^{-1}$, SCH23390 hydrochloride, #D054 Sigma-Aldrich), a dopamine D1 receptor antagonist and DL-Propranolol hydrochloride (5 mg ml$^{-1}$, #P0884, Sigma-Aldrich), a β-adrenergic receptor antagonist, were dissolved in 0.9% saline solution. All drugs were injected into aIC immediately after the post-conditioning session at a rate of 0.25 μl min$^{-1}$. A video camera was mounted above the conditioned place preference compartments and the time spent in each compartment was tracked with Debut Video Capture-NCH computer software version 5.73.

**Real-time conditioned place preference (rtCPP)**. The rtCPP apparatus (20 × 20 × 42 cm) consisted of an acrylic arena half divided with two different visual cues. One side had black/white stripes and the other side had black/white circles allowing mice freely to cross between sides. The position of the visual cues was counterbalanced in the experimental room. Mice were always free to explore the acrylic arena. During the pre-conditioning session (day 1), mice explored the arena for 10 min. The mice were conditioned in the side where they spent less time (less preferred) during the pre-conditioning session. During the 20-min conditioning sessions (day 2–4), the mice were photoactivated into VTA while they were in the conditioned side. The blue laser (473 nm) was pulsed at 20 Hz (5 ms width), 12–14 mW. ChR2 mice that spent at least 75% out of the total time during the third conditioning session in the conditioned side were included for all the experiments. During the post-conditioning session (day 5), ChR2 and eYFP mice explored the arena for 10 min without photostimulation. On the other hand, the mice injected in the BLA with NpHR3.0 (and corresponding eYFP control mice) received photoinhibition with the green laser into aIC throughout the entire post-conditioning session (day 5). The green laser (532 nM) was constantly pulsed at 12–14 mW. For all mice, extinction (days 6–8) and post-reinstatement sessions (days 10–16) were performed in the acrylic arena for 10 min. The reinstatement of 5 min (day 9) consisted of photoactivation of the VTA while mice were in the previously conditioned side, using the same stimulation parameters as the conditioning sessions. The reversal conditioning (days 11–13) was performed through photoactivation of the VTA in the unconditioned side in free-moving mice after 5 extinction sessions (days 6–10). The preference index in the conditioned side was calculated by dividing the time spent in the conditioned side by the total time of exploration.

**Microdialysis**. Dialysis procedures began by connecting the probe inlet (dialysis probes CMA/7 from CMA Microdialysis, with a 2 mm total length of membrane) to the microinfusion pump system (CMA/Microdialysis), which infused continuously Ringer solution (NaCl 144 mM, KCl 4.8 mM, MgCl$_2$ 1.2 mM, and CaCl$_2$ 1.7 mM) at a rate of 0.25 μl min$^{-1}$. The first 60 min of sampling were taken as a stabilization period. Through the microdialysis membrane, a total of six consecutive samples were collected in fractions of 16 min (4 μl) in vials containing 1 μl of an antioxidant mixture (0.25 nM ascorbic acid, Na$_2$EDTA 0.27 mM, and 0.1 M acetic acid). The stabilization period was performed in a wooden box (30 × 17 × 19 cm) with two side gates. The wooden box was inside the rtCPP apparatus or microdialysis chamber (diameter 30 cm × height 30 cm). Three samples were taken as a baseline. Once sample 3 was fully collected, the side gates were opened, and the wooden box was removed after the mouse left it. ChR2 and eYFP mice in the rtCPP apparatus were previously conditioned and sample 4 was collected during the post-conditioning session. ChR2 and eYFP naïve mice remained in the microdialysis chamber and sample 4 was collected during photoactivation of the VTA. Two more samples were recollected after the stimulus. The samples were stored at −80 °C until analyzed with the capillary electrophoresis method (see below).

**Neurotransmitters analysis**. Neurotransmitter concentration was determined as previously described[67]. Briefly, samples were derivatized with 5-furoylquinoline-3-carbaldehyde (FQ) and analyzed by a Capillary Electrophoresis system (Beckman-Coulter PACE/MDQ, Glycoprotein System) with laser-induced fluorescence detection. Derivatization reaction: 4 μl of a sample was spiked with 1 μl internal

standard (1 mg/ml, O-methyl-L-threonine), mixed with 6 μl of FQ (16.67 mM) and 2 μl of KCN (25 mM). The mixture reacted in the dark (65 °C) for 15 min and the reaction was stopped with ice. The identification of neurotransmitters is based on the retention time in a micellar electrokinetic chromatography buffer system that includes borates 35 mM, sodium dodecyl sulfate 25 mM, and 13% methanol, at pH 9.5. The samples were injected hydro-dynamically at 0.5 psi for 5 s; then the separation was performed at 20 kV. While the molecules migrate inside the capillary they move through a window where an argon-ion laser excites them at 488 nm. Then fluorescence was filtered by a band-pass interference filter at 590 nm and detected by a photomultiplier tube. The signal was translated into peaks that form an electropherogram where they can be analyzed by comparing the area under the curve (AUC) of the unknown sample with the AUC of a known internal standard. Data were analyzed using the Karat System Gold (Beckman Coulter) version 5.0. All results were converted into a percentage of baseline release (% baseline = neurotransmitter concentration × 100/mean of the three first samples).

**Immunofluorescence**. Mice were sacrificed with sodium pentobarbital (75 mg/kg), perfused with 0.9% saline solution and fixed with 4% paraformaldehyde solution. Brain was quickly removed and stored in 4% paraformaldehyde solution. Leica CM520 cryostat was used to obtain 40 μm coronal slices. Free-floating sections were incubated with rabbit polyclonal anti-TH primary antibody (Pel-Freez, Rogers, AR) dissolved in 5% bovine serum albumin buffer (NaCl 150 mM, Triton X-100 0.1%, trizma base 100 mM, pH 7.4) at a dilution of 1:1000 overnight. Sections were washed with Tris-buffered saline and incubated with goat anti-rabbit IgG conjugated to CY3 secondary antibody (Millipore, Darm-Stadt, Germany) dissolved in 5% bovine serum albumin buffer at a dilution of 1:250 for 2 h. Sections were mounted in Dako fluorescence mounting medium with 4′,6-diamidino-2-fenilindol. Immunofluorescence was observed using a ZEISS LSM 800 confocal microscope.

**Optrode recordings during photoactivation of VTA neurons**. TH-Cre mice (ChR2, $n = 11$) were injected with ChR2 bilaterally into the VTA (from Bregma (mm): −3.0 AP, ±0.60 ML, and −4.80 DV) and 3 weeks of recovery were allowed for ChR2 expression. After this time, eight mice were implanted bilaterally with optic fibers at VTA (from Bregma (mm): −3.0 AP, ±1.20 ML, and −4.30 DV, 10° angle) as described above, but in this case, an array of 16 tungsten wires was implanted unilaterally in the aIC (from Dura (mm): +1.4 AP, ±3.30 ML, −3.20 DV). Another group of mice ($n = 3$) were implanted with a homemade optrode (16 tungsten wires, 35 μm each) surrounding the optic fiber[21] in the VTA (from Dura (mm): −3.0 AP, ±0.60 ML, and −4.0 DV). Mice were allowed 1 additional week to recover. Mice were recorded in a scanner laser frequency task[21], where they received a random frequency of stimulation during 1 s on, followed by 2 s off, and the next frequency was randomly chosen. The frequencies of stimulation were: control (no stimulation), 4, 7, 20, 33, 50 Hz (5 ms pulse width), and a 1 s continuous pulse. The task was carried out for 20 min. Mice were recorded in maximum of nine consecutive sessions.

**Optrode recordings during photoinhibition of BLA → aIC fibers**. One group of TH-Cre mice (NpHR3.0/ChR2, $n = 3$) were injected with ChR2 bilaterally into the VTA (from Bregma (mm): −3.08 AP, ±0.60 ML, and −4.80 DV), NpHR3.0 bilaterally into the BLA (from Bregma (mm): −2.05 AP, ±3.40 ML, and −5.30 DV), and 3 weeks of recovery were allowed for ChR2 and NpHR expression. After that, an optrode was implanted unilaterally in the aIC (from Dura (mm): +1.4 AP, ±3.30 ML, and −3.20 DV). Another group of TH-Cre mice (NpHR3.0, $n = 5$) were injected only with NpHR3.0 bilaterally into the BLA (from Bregma (mm): −2.0 AP, ±3.40 ML, and −5.30 DV), and an optrode was implanted in aIC (from Dura (mm): +1.4 AP, ±3.30 ML, and −3.20 DV) after 3 weeks. After surgery, mice were allowed 1 additional week for recovery. Then both groups of mice were recorded and photoinhibited into aIC during an open-loop task that consisted of blocks of 1 min laser on and 1 min off for 30 min (constant pulsed at 12–14 mW and 532 nm), in seven consecutive sessions.

**Extracellular recordings**. Neural activity was recorded using a Multichannel Acquisition Processor system (Plexon, Dallas, TX) interfaced with a Med Associates conditioning side of the rtCPP to record behavioral events simultaneously. Extracellular voltage signals were first amplified ×1 by an analog headstage (Plexon HST/16o25-GEN2-18P-2GP-G1), then amplified (×1000) and sampled at 40 kHz. Raw signals were band-pass filtered from 154 to 8.8 kHz and digitalized at 12 bits resolution. Only single neurons with action potentials with a signal-to-noise ratio of 3:1 were analyzed[68]. The action potentials were isolated on-line using voltage-time threshold windows and three principal components contour templates algorithm. A cluster of waveforms was assigned to a single unit if two criteria were met: inter-spike intervals were larger than the refractory period set to 1 ms, and if a visible ellipsoid cloud composed of the 3-D projections of the first three principal component analysis of spike waveform shapes was formed. Spikes were sorted using Offline Sorter software (Plexon, Dallas, TX)[68]. Only time stamps from offline-sorted waveforms were analyzed. All electrophysiological data were analyzed using MATLAB (The MathWorks Inc., Natick, MA).

**Statistics and reproducibility**. Behavioral and microdialysis data were analyzed in GraphPad Prism 8. When the data were normally distributed according to the Kolmogorov–Smirnov test, a repeated measure two-way ANOVA was used with the Geisser–Greenhouse correction for sphericity. Bonferroni and Tukey's multiple comparisons test was performed when appropriate. Non-normally distributed data were analyzed using Mann–Whitney test. All data are showed as mean ± SEM. $\alpha$ was always set $p < 0.05$. All statistical tests used were two-tail tests. The electro-physiological analysis was performed using Matlab 2015a. In the case of electro-physiological recordings, the firing rate was compared using a nonparametric Kruskal–Wallis test followed by a Tukey–Kramer post hoc. Neurons whose firing rates were significantly different ($p < 0.05$) from the control trial during the laser "on" period at any laser frequency were categorized as modulated. The difference between firing rates during laser stimulation vs. control trials determined the sign of the modulation (i.e., increased or decreased).

For behavioral experiments, sample size was the minimum necessary to get an effect size threshold of 0.5 using the Cohen's d test dividing the difference between the means and the pooled standard deviation. Due to our previous experience in microdialysis studies, we used the minimal number of mice to get a statistical significance using nonparametric analysis. For electrophysiological recordings our sample size was based on similar previous studies and all groups were replicated independently. Data analysis was performed by blind observers, that is, mice were identified only after getting the behavioral, electrophysiological or microdialysis parameters.

**Reporting summary**. Further information on research design is available in the Nature Research Reporting Summary linked to this article.

## Data availability

The datasets and codes generated during and/or analyzed during the current study are available in the Figshare repository, https://doi.org/10.6084/m9.figshare.11418195.v1.

## Code availability

Code was written in Matlab 2015a and it is available upon request from the corresponding author.

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

## Acknowledgements

We would like to dedicate this work to Dr I.B., who unfortunately passed away very young. We would also thank Dr Rui M. Costa for the TH-Cre mice; Dr James McGaugh, and Dr Carlos Rodriguez-Ortiz for their comments on an earlier version of the manuscript; Dr Luis Rodríguez Durán for technical assistance; M.C. Oscar Giovanny Urrego Morales for helpful discussions and M.C. Gerardo Ramirez Mejia for pictures design assistance. We also thank the Unidad de Imagenología of the Instituto de Fisiología Celular, UNAM for the image processing, especially to Dr Abraham Rosas Arellano. E.G.L. is a doctoral student from Programa de Doctorado en Ciencias Biomédicas, Universidad Nacional Autónoma de México (UNAM) and received the fellowship 573989 from Consejo Nacional de Ciencia y Tecnología (CONACYT), México. This work was supported by the CONACyT grant 250870, CONACyT grant FOINS 474, and DGAPA-PAPIIT-UNAM grant IN212919.

## Author contributions

E.G-L., I.B., F.T., R.G., A.B., R.A.McD. and F.B-R. designed the research. E.G-L., I.B., and P.M-C. performed in vivo microdialysis, behavioral. and optogenetic experiments. E.G-L. and J.L-I. performed in vivo optrode electrophysiological experiments. E.G-L. and I.B. performed imaging analysis. E.G-L., R.G., A.B., R.A.McD. and F.B-R. wrote the paper. E.G-L. created the mice, boxes, and brain images. All experiments were done in Mexico City labs. All mice and boxes elements were created by E.G.-L., brain images were vectorized using a public domain sagittal brain image as template, and coronal brain pictures were modified from Allen's reference atlas of the mouse brain.

## Competing interests

The authors declare no competing interests.
