## [Peer Review File · Communications Biology]

Reviewers' comments:

Reviewer #1 (Remarks to the Author):

The manuscript by Gil-Lievana et al. examines the role of the anterior insular cortex (AIC) and, particularly, the projections from the basolateral amygdala (BLA) to the AIC in the maintenance/extinction of a conditioned place preference (CPP). To induce the CPP, the authors used photostimulation of the VTA to produce the CPP. They found that blockade of the NMDA receptors in the AIC accelerated the extinction and reduced the ability of a reinstatement procedure to restore the CPP. Similarly, they found that photoinhibition of BLA axons in the AIC produced essentially the same results. The authors place these findings in the context of reconsolidation. Overall, the findings are rather interesting and the studies appear to be well done. Considering the increased interest in the insular cortex, the manuscript is also rather timely, providing the field with new ideas about the interaction between the amygdala and insular cortex with regard to reward learning and extinction. The statistical analyses and methods are well described and I believe that a research could replicate the work. The manuscript is generally well written, though I have a few specific concerns detailed below.

1. To fully place the findings in a reconsolidation context, the authors would need to do additional studies, such as giving the manipulation without the CPP test. Moreover, there appears to be a problem in comparing the NMDA receptor blockade to the inhibition of BLA inputs to the AIC. For the NMDA receptor blockade, AP5 was administered into the AIC after the post-conditioning test and was found to be effective for enhancing extinction. In contrast, inhibition of BLA inputs to the AIC was only effective when performed during the test and not after. If the idea is that the AP5 was blocking glutamatergic signaling that included glutamate released from the BLA terminals, these findings are not consistent with that. Indeed, one would expect both manipulations to work equally effectively. The authors should address this discrepancy.
2. My own view is that this calls into question whether these are truly "reconsolidation" results. One possibility is that BLA inputs to the AIC may be important for "maintaining" the memory (as the title suggests) and that disrupting this enables extinction to forward more quickly. This is quite different, however, than demonstrating reconsolidation. As it is, I would suggest that the authors re-frame their findings and only acknowledge that reconsolidation is a *possible* explanation of the results.
3. The authors appear to have used both males and females but, unless I missed it, do not comment on whether any sex differences (trends or otherwise) were observed. They should do so.
4. Figure 4 has panels b and c reversed from the text.

Minor:

Page 13, line 295 – should read "labilized"

Reviewer #2 (Remarks to the Author):

In this study, the authors focused on the role of the anterior insular cortex (AIC) and its input from the basolateral amygdala (BLA) in mediating the association between contextual information and reward. The authors used various techniques to show that stimulation of tyrosine hydroxylase (TH)-expressing neurons in the ventral tegmental area (VTA) is sufficient to induce conditioned place preference (CPP) and the retrieval of such memories require neurotransmission in the AIC. Further, intra-AIC microinjections of the glutamate receptor antagonist NMDA facilitate extinction of the CPP memory, as does inhibition of BLA projections to the AIC.

While I believe this manuscript has many strengths, it also suffers from several weaknesses that need to be addressed before being considered for publication.

1. I would recommend that the authors restructure their Introduction to better frame their study. Currently, the introduction describes the role of the AIC and BLA-AIC projects in the context of addiction. None of the experiments, however, involve any drug administration or manipulation. I agree that these experiments will indeed inform our understanding of the neurobiology underlying drug addiction, but I do not think that two paragraphs worth of detail is necessary.
2. Given the methods and results, the authors should introduce memory mechanisms and how the AIC and BLA-AIC projections may be involved.
3. The authors should provide a stronger rationale for the approach they chose when designing the experiment. For example, why did the authors choose to use VTA stimulation rather than a drug stimulus to promote CPP?
4. I would recommend that the authors review the paper from Bernard Balleine's lab that focuses on BLA interactions with AIC (e.g., Parkes and Balleine, 2013)—it might be useful in the Introduction or Discussion.
5. There is no need to discuss the results in such detail in the Introduction.
6. The experimental group labeling is extremely confusing in the text and in the figures. Instead of Chr2-, for example, it would be easier to understand that this is the control vector if the authors used eYFP. Similarly, in Figure 1d, Chr2/extinction and Chr2/reinstatement imply that one group is stimulated during extinction and the other during reinstatement, which is not the case. Please use more intuitive group names for ease of reading.
7. In the Methods section, it is difficult to understand which details pertain to which manipulation. Perhaps it would be helpful to separate out the Methods by "experiment" or use subsections. For example, in the photoinhibition section, it would be easier to read and understand if the authors had separate subsections for 1. The Chr2 and eNpHR3.0 injections + optode implant vs. 2. eNpHR3.0 + optode. It could be framed either as separate experiments or just have subheadings; in either case, it will allow readers to follow the methods more clearly.
8. What is the timing/durations of the behavioral sessions? In particular, are the 3 conditioning sessions on the same day or do they occur across 3 days? What is the interval between the last conditioning session and the post-conditioning session? What is the interval between the post-conditioning session and the first extinction session? Do all extinction sessions occur on the same day or do they occur across 5 days? These are important details as they can have a significant impact on the interpretation of the results.
9. I was slightly confused about the experimental design for experiments reflected in Figure 3. It was not clear whether VTA stimulation occurred during a CPP session (causing CPP) or that VTA stimulation occurred alone (i.e., not in the CPP training context). The authors loosely equate VTA stimulation with conditioning, but it is not clear whether conditioning is actually occurring—if VTA stimulation occurs during CPP training, I would agree that this is conditioning. If it is just VTA stimulation alone, I would argue that no conditioning is occurring and that the authors need to adjust the wording. I would also argue that, if the authors are trying to claim that these neurotransmitters are being released/involved during reconsolidation, you need to actually condition the animals (VTA stimulation in the CPP session).
10. I think that the title is misleading: the glutamatergic BLA-AIC circuit experiment was only one small part of the study. It mostly focused on the INS and the neurochemistry within. I would recommend changing the title to more broadly capture the results in the manuscript.
11. I do not think that "contextual reward" is the best way to describe what is being modeled by CPP. The association between context and VTA stimulation (which presumably is reward) is a more accurate description.
12. Is it possible for the authors to dissociate whether the effects of the various manipulations are affecting memory processes, as the authors suggest, or the reinforcing properties of the context imbued by VTA stimulation?
13. Did the authors sample neurotransmitter release in the AIC during the first extinction session?
14. There are several typos throughout the manuscript. For example,

- a. Mistake in caption for Figure 5—the figure shows A, B and C, but the caption describes A, C and D.
- b. On page 9, in the NMDA infusion section, the authors begin the paragraph by stating that they are going to characterize the role of dopamine and noreadrenaline in the AIC, but later in that same paragraph, they describe a pharmacological approach targeting the glutamatergic system. Did the authors mean glutamate in the beginning of the paragraph rather than dopamine and noreadrenaline? Could the authors clarify this discrepancy?
- c. I do not think “labialize” is actually a word (or at least word related to memory). Please re-word (pg. 13).

Reviewer #3 (Remarks to the Author):

This paper by Gil-Lievana et al. investigates the role of the BLA-aIC projection in the maintenance of rewarding contextual memories. The authors employ optogenetic stimulation of the TH+ neurons of the VTA to induce real-time conditioned place preference (rtCPP) and to establish reward-based contextual memories in mice.

The paper is subdivided into five major sections: 1) the authors first show how optogenetic stimulation of the VTA TH+ neurons induces robust rewarding context memories. 2) An assessment of how photoactivation of the VTA TH+ neurons modulates neuronal responses in the anterior insular cortex (aIC). 3) An assessment of glutamate, noradrenalin and dopamine release during rtCPP memory retrieval. 4) Intra-aIC infusions of dopamine, noradrenalin and glutamate receptor antagonists after rtCPP retrieval and assessment of extinction. 5) photoinhibition of BLA-aIC pathway during rtCPP extinction.

Overall, my impression is that the story would have the potential to be a valid contribution yet at this point it lacks crucially in clarity and straightforwardness. It is hard to follow the logic of the experimental line-up. It seems as if titration experiments, control experiments and important results are intermingled, making it hard to understand the take home messages. Yet, the manuscript addresses an important question, employs many different and adequate techniques to tackle this question, and contains some important results. Therefore, my overall recommendation would be a thorough revision mainly of the way the paper is presented and the data are lined up.

Major issues:

- Lines 156 – 167 and Fig. 2b (left) are titration experiments aimed at testing the VTA stimulation protocol – this comes after Fig. 1 where a stimulation frequency of 20 Hz has already been chosen and shown to trigger rtCPP – this does not make much sense to me. I'd suggest to put this titration into the supplement and mention it before the Fig1 rtCPP results. The same inconsistency is the summary sentence in line 174-176, where the conclusion is that 20 Hz was used – but the behavior is already shown before this in Fig. 1.
- Lines 150-154 speak about TH-positive axon terminals in aIC – this should go with the VTA-aIC pathway characterization starting in line 167

Fig.2

- Fig. 2b right and 2c right seem to show that VTA stimulation does not evoke much change in aIC firing. Since it seems that the authors report both, stimulation activated and inhibited neurons it would make sense to show these responses separately and quantitatively (similar to Fig. 6?). What was the threshold to call a neuron activated / inhibited? Even though the quantifications in d, e, f show some modulation these are not shown with error bars or statistical measures. Also, it is not clear what the threshold for 'modulation' was. I did not find a threshold or measure for categorizing 'modulation' in the methods either. Neurons in VTA (Fig. 2b lower left) seem to react to photostimulation already

before the laser onset (or is this a plotting issue?).

Lines 189 and following

- The paragraph starts stating: 'we decided to pharmacologically characterize the role of dopamine and noradrenaline receptors in the aIC on retrieval and extinction' (lines 191-193). Then, they write 'we thus administered bilateral injections of AP5' (line 200). This is totally inconsistent. Why now AP5 while the whole Fig. 3 showed post-conditioning increases of not only glutamate but also NA and DA and they themselves state they want to test roles of NA and DA?
- While it is nice to see that intra-aIC injections of AP5 post-conditioning accelerates extinction – these results come totally out of the blue – why then bother and go through all the pain of microdialysis in Fig. 3? – only in line 206 the authors then resolve this issue (partially and inadequately) mentioning in brief that SCH23390 or propranolol had no effect. The authors have to keep an order in presenting the results that remains penetrable for the reader. They also do not mention in the main text at all what SCH23390 is, neither do they explain what propranolol is – nor why they used it. Only the informed reader knows what these drugs do – it should not be assumed that the common reader of this journal know what SCH23390 does! This paragraph belongs – properly explained BEFORE the AP5 and CNQX results (at least in the current format). I would even suggest to present all the results side-by-side in Fig 4 (include data from S1 b and c - only left panels). I believe that this would be more consistent since data in Fig. 3 are presented also for NA, DA and glutamate – so it would become easier to follow for the reader.
- The second problem with the AP5 experiment is that the authors themselves cite papers that already showed that AP5 accelerates extinction – what is the novelty then here? Why is the AP5 result in this paper novel and important? What are the main differences to the earlier studies?

BLA-aIC photoinhibition

- In the last section of the paper the authors then jump to the BLA-aIC pathway. Why? While it is true that there were earlier studies it is a hard shift of direction and question.
- When exactly were the photoinhibitions done – during the post-conditioning or when done after post-conditioning. Which parameters?
- It seems very odd that even after reinstatement to a similar level the BLA-aIC inhibited group is again strongly facilitated in extinction, could this be due to opsin-cross-talk?
- An important control would be to demonstrate whether BLA-aIC inhibition itself has a valence in the rtCPP task
- I do not find the opsin-cross-talk control convincing. At least according to excitation spectra of the opsins they should activate CHR2 expressed in VTA terminals with the green light and also inhibit BLA terminals with the blue light. It doesn't help that I couldn't find the exact wavelength utilized but maybe I oversaw the mentioning?
- Implantation sites and viral spreads have to be shown for all individuals included in the analyses.

Minor comments:

- the model organism, in this case mice, should be mentioned in the abstract.
- Stimulation frequency utilized for rtCPP should be mentioned in the main text and figure, not only methods
- Line 247 typo: BLA instead of 'BIA'
- Figure 5 – panel descriptions in the text are wrong, e.g. line 237 refers to Fig. 5d which does not exist (should be Fig. 5c?)

General:

- the methods section should be extended with more detail throughout to really understand all parameters of stimulation and the timing of different experiments etc.

RESPONSES TO REVIEWER

Reviewer 1

Comment: The manuscript by Gil-Lievana et al. examines the role of the anterior insular cortex (AIC) and, particularly, the projections from the basolateral amygdala (BLA) to the AIC in the maintenance/extinction of a conditioned place preference (CPP). To induce the CPP, the authors used photostimulation of the VTA to produce the CPP. They found that blockade of the NMDA receptors in the AIC accelerated the extinction and reduced the ability of a reinstatement procedure to restore the CPP. Similarly, they found that photoinhibition of BLA axons in the AIC produced essentially the same results. The authors place these findings in the context of reconsolidation. Overall, the findings are rather interesting, and the studies appear to be well done. Considering the increased interest in the insular cortex, the manuscript is also rather timely, providing the field with new ideas about the interaction between the amygdala and insular cortex with regard to reward learning and extinction. The statistical analyses and methods are well described, and I believe that a research could replicate the work. The manuscript is generally well written, though I have a few specific concerns detailed below.

1. To fully place the findings in a reconsolidation context, the authors would need to do additional studies, such as giving the manipulation without the CPP test. Moreover, there appears to be a problem is comparing the NMDA receptor blockade to the inhibition of BLA inputs to the AIC. For the NMDA receptor blockade, AP5 was administered into the AIC after the post-conditioning test and was found to be effective for enhancing extinction. In contrast, inhibition of BLA inputs to the AIC was only effective when performed during the test and not after. If the idea is that the AP5 was blocking glutamatergic signaling that included glutamate released from the BLA terminals, these findings are not consistent with that. Indeed, one would expect both manipulations to work equally effectively. The authors should address this discrepancy.

Response: We want to thank the reviewer's comments for improving our manuscript. We observed that the post-retrieval blockade of the glutamatergic NMDA receptors with AP5 accelerates the extinction of rtCPP. However, inhibition of BLA terminals to the aIC post-retrieval did not accelerate rtCPP extinction. Interestingly, when the BLA terminals in the aIC were inhibited during retrieval, the post-conditioning rtCPP scores were not affected, while 24 hours later, the mice that received the photoinhibition showed an accelerated extinction rate. It is possible that the glutamate release in the aIC necessary for the maintenance of the rtCPP memory not only comes from the BLA, but other structures could contribute with the glutamatergic activity in the aIC after the retrieval. Although the similarities of the cellular mechanisms involved in the photoinhibition of glutamatergic terminals and the glutamatergic receptor antagonists remain to be determined, the difference observed between optogenetic inhibition of the BIA terminals in the aIC and the pharmacological inhibition of the NMDA receptors in the aIC could be also explained due to that optogenetic inhibition of BLA terminals in the aIC during retrieval unleashed similar molecular events as the blockade of the AP5 receptor after the retrieval. In this regard, we are currently evaluating whether photoinhibition

of the BLA-aIC terminals blocks the release of glutamate that affects the cellular events necessary to reconsolidate (update) context reward associations, in a similar way than we have reported with AP5 (please see, Osorio-Gomez et al., 2017) (please see, page 11, lines 256-266).

2. My own view is that this calls into question whether these are truly “reconsolidation” results. One possibility is that BLA inputs to the AIC may be important for “maintaining” the memory (as the title suggests) and that disrupting this enables extinction to forward more quickly. This is quite different, however, than demonstrating reconsolidation. As it is, I would suggest that the authors re-frame their findings and only acknowledge that reconsolidation is a *possible* explanation of the results.

Response: We agree with the reviewer, the reconsolidation phenomenon is one possibility. We hypothesized that the optogenetic inhibition of the BLA projections in the aIC impairs the reconsolidation process of the rewarding contextual memory, but we can not rule out that alternative mechanisms involved in the maintenance of the rewarding memory, could be disrupted and be reflected in an accelerated extinction rate (please see, page 11, lines 264-268).

3. The authors appear to have used both males and females but, unless I missed it, do not comment on whether any sex differences (trends or otherwise) were observed. They should do so.

Response: No difference was observed between mice’s sex in any of the different phases analyzed in the rtCPP or in the neurotransmitter profile, which is reflected in the small standard error of the mean in the different groups.

4. Figure 4 has panels b and c reversed from the text.

Response: We apologize for this oversight, it is now fixed.

5. Minor: Page 13, line 295 – should read “labilized”

Response: We changed the word: labialized to labile page 10, line 233.

Reviewer 2

In this study, the authors focused on the role of the anterior insular cortex (AIC) and its input from the basolateral amygdala (BLA) in mediating the association between contextual information and reward. The authors used various techniques to show that stimulation of tyrosine hydroxylase (TH)-expressing neurons in the ventral tegmental area (VTA) is sufficient to induce conditioned place preference (CPP) and the retrieval of such memories require neurotransmission in the AIC. Further, intra-AIC microinjections of the glutamate receptor antagonist NMDA facilitate extinction of the CPP memory, as does inhibition of BLA projections to the AIC.

While I believe this manuscript has many strengths, it also suffers from several weaknesses that need to be addressed before being considered for publication.

1. I would recommend that the authors restructure their Introduction to better frame their study. Currently, the introduction describes the role of the AIC and BLA-AIC projects in the context of addiction. None of the experiments, however, involve any drug administration or manipulation. I agree that these experiments will indeed inform our understanding of the neurobiology underlying drug addiction, but I do not think that two paragraphs' worth of detail is necessary.

Response: We are grateful to the reviewer for these comments to strengthen our manuscript. We restructured the introduction. In accordance with the reviewer's suggestion, we have modified the order of the introduction section. First, we describe the role of the aIC in addiction in humans and rodents. After, the relationship between drugs of abuse and brain's reward circuitry driven by dopaminergic and glutamatergic neurotransmission. In addition, we have discussed how glutamatergic synaptic plasticity in addiction shares the same type of molecular changes involved in learning and memory processes. Finally, we describe how glutamatergic communication between BLA and aIC participates in the consolidation of several kinds of memories (please see, paragraphs 1-2 on pages 3).

2. Given the methods and results, the authors should introduce memory mechanisms and how the AIC and BLA-AIC projections may be involved.

Response: We add a paragraph about of memory mechanisms and how the AIC and BLA-AIC projections may be involved (please see, page 3, lines 65-69).

3. The authors should provide a stronger rationale for the approach they chose when designing the experiment. For example, why did the authors choose to use VTA stimulation rather than a drug stimulus to promote CPP?

Response: Previous studies have demonstrated that activation of VTA dopaminergic neurons is enough to produce a conditioning place preference (CPP) (please see, Tsai et al., 2009). The optogenetic stimulation of VTA allows us to have better control of the temporal and spatial activity of the dopaminergic neurons. Consequently, by using this model we are able to control the interaction of the VTA with other relevant brain circuits participating in the rtCPP. In

contrast, drug-induced CPP has some disadvantages, such as the nonspecific effects on brain circuits by systemic injections of drugs (please see, page 9, lines 201-206).

4. I would recommend that the authors review the paper from Bernard Balleine's lab that focuses on BLA interactions with AIC (e.g., Parkes and Balleine, 2013)—it might be useful in the Introduction or Discussion.

Response: We reviewed the paper and we considered relevant for the introduction section (please see, page 3, line 71).

5. There is no need to discuss the results in such detail in the Introduction.

Response: Following the Reviewer's suggestion, we now briefly summarized the results in the introduction (please see, page 4, lines 76-93).

6. The experimental group labeling is extremely confusing in the text and in the figures. Instead of ChR2-, for example, it would be easier to understand that this is the control vector if the authors used eYFP. Similarly, in Figure 1d, ChR2/extinction and ChR2/reinstatement imply that one group is stimulated during extinction and the other during reinstatement, which is not the case. Please use more intuitive group names for ease of reading.

Response: We have changed the group labeling to ChR2 (cation channel), NpHR3.0 (chloride ions pump) and eYFP (control vector) (please see, figure 1-6 on pages 25-30). We also changed the group labeling ChR2/With reinstatement and ChR2/Without reinstatement (please see, page 26, figure 2d).

7. In the Methods section, it is difficult to understand which details pertain to which manipulation. Perhaps it would be helpful to separate out the Methods by "experiment" or use subsections. For example, in the photoinhibition section, it would be easier to read and understand if the authors had separate subsections for 1. The ChR2 and eNpHR3.0 injections + optode implant vs. 2. eNpHR3.0 + optode. It could be framed either as separate experiments or just have subheadings; in either case, it will allow readers to follow the methods more clearly.

Response: We have added subheadings to separate both experiments, in NpHR3.0/ChR2 and NpHR3.0 (please see, page 17, lines 404-414).

8. What is the timing/durations of the behavioral sessions? In particular, are the 3 conditioning sessions on the same day or do they occur across 3 days? What is the interval between the last conditioning session and the post-conditioning session? What is the interval between the post-conditioning session and the first extinction session? Do all extinction sessions occur on the same day or do they occur across 5 days? These are important details as they can have a significant impact on the interpretation of the results.

Response: Now we have indicated the experimental days in the results section and in the diagram of the figures to facilitate the interpretation of the results (please see, page 14, lines

330-341 and insets on figures 2, 3, 4, 5, S2 on pages 26-32).

9. I was slightly confused about the experimental design for experiments reflected in Figure 3. It was not clear whether VTA stimulation occurred during a CPP session (causing CPP) or that VTA stimulation occurred alone (i.e., not in the CPP training context). The authors loosely equate VTA stimulation with conditioning, but it is not clear whether conditioning is actually occurring—if VTA stimulation occurs during CPP training, I would agree that this is conditioning. If it is just VTA stimulation alone, I would argue that no conditioning is occurring and that the authors need to adjust the wording. I would also argue that, if the authors are trying to claim that these neurotransmitters are being released/involved during reconsolidation, you need to actually condition the animals (VTA stimulation in the CPP session).

Response: We reorganized the inset diagram to indicate the days in which rtCPP was performed (please see, page 27, figure 3), we clarified the temporal window when microdialysis was done. First, we tested the neurochemical responses in the aIC under photoactivation of the VTA in non-conditioned mice and second during the rtCPP memory retrieval (post-conditioning) session in mice previously conditioned (please see, page 6, lines 135-137).

10. I think that the title is misleading: the glutamatergic BLA-AIC circuit experiment was only one small part of the study. It mostly focused on the INS and the neurochemistry within. I would recommend changing the title to more broadly capture the results in the manuscript.

Response: Although, we agree with the reviewer that the BLA-aIC circuit is the last part of the experiments. In this paper, we tried to analyze the role of IC glutamatergic activity, as well as its possible origins in the circuits involved in the maintenance of rtCPP. We believe that the study of cortical glutamatergic circuits involved in the maintenance of rewarding contextual conditioning is essential to shedding light on the brain mechanisms involved in the maintenance of drug addictions.

11. I do not think that “contextual reward” is the best way to describe what is being modeled by CPP. The association between context and VTA stimulation (which presumably is reward) is a more accurate description.

Response: In our experiments, the side associated with VTA stimulation becomes a positive reinforcer and increases the likelihood of mice to spend more time in the reinforced side. For this reason, we decided to use the term "contextual reward memory" that we believe best describes our behavior model.

12. Is it possible for the authors to dissociate whether the effects of the various manipulations are affecting memory processes, as the authors suggest, or the reinforcing properties of the context imbued by VTA stimulation?

Response: Yes, the effect we observe in maintenance is affected by memory processes and not by the context imbued by VTA stimulation, since all experimental pharmacological and photoinhibition manipulations were performed during and after the retrieval (post-test) and not

during VTA stimulation.

13. Did the authors sample neurotransmitter release in the AIC during the first extinction session?

Response: Yes, *in vivo* microdialysis was done on the post-conditioning session, in a way, the first extinction session.

14. There are several typos throughout the manuscript. For example,
a. Mistake in caption for Figure 5—the figure shows A, B and C, but the caption describes A, C and D.

Response: We apologize for the mistakes, they are already fixed in the text.

b. On page 9, in the NMDA infusion section, the authors begin the paragraph by stating that they are going to characterize the role of dopamine and noreadrenaline in the AIC, but later in that same paragraph, they describe a pharmacological approach targeting the glutamatergic system. Did the authors mean glutamate in the beginning of the paragraph rather than dopamine and noreadrenaline? Could the authors clarify this discrepancy?

Response: Thank you for the observation. We reorganized the NMDA infusion section, first, the results of catecholamines are described, and then we continue with the glutamatergic system results (please, see pages 6-7, lines 148-163).

c. I do not think “labialize” is actually a word (or at least word related to memory). Please reword (pg. 13).

Response: We changed the word: labialized to labile now on page 10, line 233.

Reviewer 3

This paper by Gil-Lievana et al. investigates the role of the BLA-aIC projection in the maintenance of rewarding contextual memories. The authors employ optogenetic stimulation of the TH+ neurons of the VTA to induce real-time conditioned place preference (rtCPP) and to establish reward-based contextual memories in mice.

The paper is subdivided into five major sections: 1) the authors first show how optogenetic stimulation of the VTA TH+ neurons induces robust rewarding context memories. 2) An assessment of how photoactivation of the VTA TH+ neurons modulates neuronal responses in the anterior insular cortex (aIC). 3) An assessment of glutamate, noradrenalin and dopamine release during rtCPP memory retrieval. 4) Intra-aIC infusions of dopamine, noradrenalin and glutamate receptor antagonists after rtCPP retrieval and assessment of extinction. 5) photoinhibition of BLA-aIC pathway during rtCPP extinction.

Overall, my impression is that the story would have the potential to be a valid contribution yet at this point it lacks crucially in clarity and straightforwardness. It is hard to follow the logic of the experimental line-up. It seems as if titration experiments, control experiments and important results are intermingled, making it hard to understand the take home messages. Yet, the manuscript addresses an important question, employs many different and adequate techniques to tackle this question, and contains some important results. Therefore, my overall recommendation would be a thorough revision mainly of the way the paper is presented, and the data are lined up.

Major issues:

1. Lines 156 – 167 and Fig. 2b (left) are titration experiments aimed at testing the VTA stimulation protocol – this comes after Fig. 1 where a stimulation frequency of 20 Hz has already been chosen and shown to trigger rtCPP – this does not make much sense to me. I'd suggest to put this titration into the supplement and mention it before the Fig1 rtCPP results. The same inconsistency is the summary sentence in line 174-176, where the conclusion is that 20 Hz was used – but the behavior is already shown before this in Fig. 1.

Response: We appreciate the comments and insights to improve our manuscript. As suggested by the reviewer, we modify the order of the figures. The result set that describes the validation of the model is now Figure 1 and the result set of rtCPP is now Figure 2 (please, see pages 4-6, lines 95-133 and figures 1-2 on pages 25,26).

2. Lines 150-154 speak about TH-positive axon terminals in aIC – this should go with the VTA-aIC pathway characterization starting in line 167

Response: To clarify the question raised by the reviewer: we added “inputs from VTA into aIC” (on page 4, line 96). To corroborate the presence of a dopaminergic VTA→aIC pathway we first characterized the expression of eYFP protein in VTA neurons to the aIC of ChR2 mice. After that, we characterized the functional modulation of the VTA→aIC circuit, by performing electrophysiological recordings in VTA or aIC in freely moving naïve mice, during photostimulation of VTA TH+ neurons (please, see page 4-5, lines 96-103).

3. Fig. 2b right and 2c right seem to show that VTA stimulation does not evoke much change in aIC firing. Since it seems that the authors report both, stimulation activated and inhibited neurons it would make sense to show these responses separately and quantitatively (similar to Fig. 6?). What was the threshold to call a neuron activated / inhibited? Even though the quantifications in d, e, f show some modulation these are not shown with error bars or statistical measures. Also, it is not clear what the threshold for 'modulation' was. I did not find a threshold or measure for categorizing 'modulation' in the methods either. Neurons in VTA (Fig. 2b lower left) seem to react to photostimulation already before the laser onset (or is this a plotting issue?).

Response: Thank you for your thoughtful comments. As suggested by the Reviewer, we have modified the previous Figure 2 now Figure 1 with the same style than Figure 6. Figure 1 now had a heatmap with the neuronal responses split as a function of increase or decrease responses (please see, page 25). Moreover, we also include a new figure, the supplementary Figure S1 (please see, page 31), where it is plotted the percentage of inhibited or activated neurons for each stimulation frequency and brain region. This figure also displays the analysis of spike-laser coherence.

Regarding your question about the threshold, we apologize for this missing information. We use a Kruskal-Wallis test to determine a modulation, we have now written in Methods the following:

“In the case of electrophysiological recordings, the firing rate was compared using a non-parametric Kruskal-Wallis test followed by a Tukey-Kramer post hoc. Neurons which firing rates, during the laser “on” period was significantly different ($p < 0.05$), at any laser frequency, relative to the activity in the control trial was considered as modulated. The difference between firing rates during laser stimulation vs. control trials determined the sign of the modulation (i.e., increased or decreased).” (please see, page 18-19, lines 437-442)

Regarding your question: “Even though the quantifications in d, e, f show some modulation these are not shown with error bars or statistical measures.” Note that there are no error bars, because they are the percentage of total neurons recorded (for this information see now Supplementary Figure 1 on page 31).

Finally, relative to your question: “Neurons in VTA (Fig. 2b lower left) seem to react to photostimulation already before the laser onset (or is this a plotting issue?).” Thank you for noticing this point. There is no activation preceding laser stimulation the reason for this is that for visualization purposes, the responses were smoothed, to minimize this issue, we have now used a smaller bin size and less smoothing.

4. The paragraph starts stating: ‘we decided to pharmacologically characterize the role of dopamine and noradrenaline receptors in the aIC on retrieval and extinction’ (lines 191-193). Then, they write ‘we thus administered bilateral injections of AP5’ (line 200). This is totally inconsistent. Why now AP5 while the whole Fig. 3 showed post-conditioning increases of not only glutamate but also NA and DA and they themselves state they want to test roles of NA and DA?

- While it is nice to see that intra-aIC injections of AP5 post-conditioning accelerates extinction

– these results come totally out of the blue – why then bother and go through all the pain of microdialysis in Fig. 3? – only in line 206 the authors then resolve this issue (partially and inadequately) mentioning in brief that SCH23390 or propranolol had no effect. The authors have to keep an order in presenting the results that remains penetrable for the reader. They also do not mention in the main text at all what SCH23390 is, neither do they explain what propranolol is – nor why they used it. Only the informed reader knows what these drugs do – it should not be assumed that the common reader of this journal know what SCH23390 does! This paragraph belongs – properly explained BEFORE the AP5 and CNQX results (at least in the current format). I would even suggest presenting all the results side-by-side in Fig 4 (include data from S1 b and c - only left panels). I believe that this would be more consistent since data in Fig. 3 are presented also for NA, DA and glutamate – so it would become easier to follow for the reader.

Response: Thank you for the observation. We reorganized the NMDA infusion section, first we described the results of catecholamines and after the results of glutamatergic antagonists injections (please, see page 6, lines 148-163). In addition, we included the results using of SCH23390 and propranolol in figure 4 (please, see on page 28).

5. The second problem with the AP5 experiment is that the authors themselves cite papers that already showed that AP5 accelerates extinction – what is the novelty then here? Why is the AP5 result in this paper novel and important? What are the main differences to the earlier studies?

Response: We designed a systematic method to study the cellular mechanisms underlying the reward contextual memory. First, during memory retrieval, we found a strong release of glutamate, noradrenaline, and dopamine in the aIC. Therefore, we administrated post-retrieval injections of receptor antagonists of the released neurotransmitters to interfere with the rtCPP reconsolidation processes. In this regard, it has been demonstrated that post-trial injections of AP5 (an NMDA-receptor antagonist) interfere with the reconsolidation process in different learning tasks (Santoyo-Zedillo et al., 2014; Osorio-Gómez et al., 2016; 2017). Thus, we observed, that only the NMDA receptor blockade in the aIC immediately after the retrieval session resulted in a reliable acceleration of rtCPP extinction and impairing the rtCPP reinstatement. Furthermore, we demonstrated that photoinhibition of the glutamatergic BLA→aIC afferent fibers during retrieval induces an accelerated rtCPP extinction and impaired reinstatement. Finally, our research focuses on rewarding contextual memories, whilst most of the previous studies focused on taste and object recognition memory.

BLA-aIC photoinhibition

6. In the last section of the paper the authors then jump to the BLA-aIC pathway. Why? While it is true that there were earlier studies it is a hard shift of direction and question.

Response: We modified the description of the role of the BLA→aIC pathway on the acceleration of rtCPP extinction paragraph in the results section. Now, we emphasize on the possible glutamatergic circuits with the aIC that have been identified in memory consolidation. The glutamate BLA→IC circuit has been highlighted as an important one to modify the plasticity of IC-related memory consolidation (please see page 7, lines 165-170).

7. When exactly were the photoinhibitions done – during the post-conditioning or when done after post-conditioning. Which parameters?

Response: Both cases are correct, one experimental group was photoinhibited during the post-conditioning session and other group was photoinhibited immediately after of the post-conditioning session (please see, page 8, lines 173).

8. It seems very odd that even after reinstatement to a similar level the BLA-aIC inhibited group is again strongly facilitated in extinction, could this be due to opsin-cross-talk?

Response We do not believe that the results of the photoinhibition of the BLA-aIC pathway are due to any crosstalk, since the pharmacological results using the NMDA AP5 receptor antagonist are very similar to those using the optogenetic approach. After reinstatement, both groups, ChR2/AP5 and ChR2/NpHR3.0 have similar higher extinction scores compared to their respective controls (please see figure 4c on page 28 and figure 5c on page 29). We believe that these results let us rule out an opsin-cross talk effect.

9. An important control would be to demonstrate whether BLA-aIC inhibition itself has a valence in the rtCPP task

Response: This is an interesting question, although we do not ask directly about this possibility, we know that during the BLA-aIC photoinhibition the performance of the mice was not modified. If there was any modification due to change in the valence by the photostimulation, it should be reflected immediately at the time of the stimulation session (please see, figure 5c on page 29).

10. I do not find the opsin-cross-talk control convincing. At least according to excitation spectra of the opsins they should activate CHR2 expressed in VTA terminals with the green light and also inhibit BLA terminals with the blue light. It doesn't help that I couldn't find the exact wavelength utilized but maybe I oversaw the mentioning?

Response: The wavelength of the lasers used for the activation of each opsin was described in the methods section. We used blue laser at 473 nm to excite the ChR2 opsin and green laser at 532 nm to excite the NpHR3.0 opsin. It has been reported that 532 nm excites only a small fraction of ChR2 opsins, and on the other hand, 473 nm only excites a small fraction of NpHR3.0 opsins. Since animals with an opsin showed electrophysiological scores similar to animals with two opsins, the opsin-crosstalk, although possible, we believed this would have minimal effects on the results.

11. Implantation sites and viral spreads have to be shown for all individuals included in the analyses.

Response: We added diagrams of virus infection and diffusion into VTA and BLA anteroposterior axis (please see, Figure 1a on page 25 and Figure 6a on page 30).

Minor comments:

- the model organism, in this case mice, should be mentioned in the abstract.

Response: We appreciate your comments. The abstract has been modified, and the animal model has been included in the abstract section (please see, page 2, line 34).

- Stimulation frequency utilized for rtCPP should be mentioned in the main text and figure, not only methods

Response: We already added the stimulation frequency description in the main text and in the figure caption. (please see, page 5, line 115; page 8, line 176; page 26, line 708; page 30, line 791).

- Line 247 typo: BLA instead of 'BIA'

Response: We apologize for the mistake; it is already fixed in the text.

- Figure 5 – panel descriptions in the text are wrong, e.g. line 237 refers to Fig. 5d which does not exist (should be Fig. 5c?)

Response: We apologize for this oversight, it is now fixed.

General:

- the methods section should be extended with more detail throughout to really understand all parameters of stimulation and the timing of different experiments etc.

Response: We have added all the requested details in the Methods section.

REVIEWERS' COMMENTS:

Reviewer #1 (Remarks to the Author):

The authors have adequately addressed most of the concerns. However, I have two points:

Although the authors' response indicates that there were no differences between males and females, I cannot find this statement anywhere in the text. It should be there for the reader.

Additionally, I would suggest a reader carefully go through the manuscript, as there are numerous grammatical issues throughout.

Reviewer #2 (Remarks to the Author):

The authors thoroughly addressed my questions and comments and I am satisfied with the current version.

Reviewer #3 (Remarks to the Author):

The authors have addressed all reviewer comments comprehensively and in my opinion the paper is much improved and very interesting. I fully support publication at this point and congratulate the authors.

I have two very minor comments which I believe could help the accessibility of the paper:

- 1) The data now shown in Suppl. Fig. 1a and b could be included in Fig. 1 to show how many neurons in the aIC are modulated since it is very hard to see from the data plotted in Fig. 1e and f.
- 2) I would suggest adding to the labels in the graphs of Fig. 4b and c the information ('dopamine D1 receptor antagonist (SCH23390)'; ' β -adrenergic receptor antagonist (Propranolol)'; 'NMDA receptor antagonist (AP5)'; 'AMPA receptor antagonist (CNQX)') to make it easier to understand.

RESPONSES TO REVIEWERS' COMMENTS

We want to thank the reviewer's comments for improving our manuscript.

1. Include a statement that there were no differences b/t males and females.

Response: we wrote in animals' section the following statement:

"No differences were found between male and female TH-Cre mice in any of the experiments performed" (please see, page 12, lines 275-276).

2. Correct all grammatical typos, we suggest to do editorial peer review (see more details in the attached document).

Response: we apologize for the mistakes, they are already fixed in the text.

3. Include Supp Fig 1a and b in Fig 1.

Response: we included the supplementary figure 1a and 1b in the Figure 1 (please see, page 5, lines 112-115 and page 25-26, lines 694-697).

4. Add labels in the graphs of Fig4b and c

Response: we modified the group labels dopamine D1 receptor antagonists (SCH23390) and β -adrenergic receptor antagonists (Propranolol) in the figure 4b. Also, glutamatergic NMDA receptor antagonist (AP5) and glutamatergic AMPA receptor antagonist (CNQX) in the figure 4c (please see, page 28).

5. Please review the attached document and address the comments and queries therein, which include but is not limited to, changing "Data Analysis and statistics" to "Statistics and Reproducibility"

Response: we changed the subtitle "Data Analysis and statistics" to "Statistics and Reproducibility" (please see, page 18, line 430).

6. Overlay dot plots over bar graphs and avoid using stacked bar graphs

Response: we edited the graphs overlaying dot plots over bar graphs (please see, page 27, figure 3).